# A dynamic generative model can extract interpretable oscillatory components from multichannel neurophysiological recordings

**Proloy Das[1]\*, Mingjian He[1,2], Patrick L Purdon[1,3]**

[1]Department of Anesthesiology, Perioperative and Pain Medicine, Stanford University, Stanford, United States; [2]Department of Psychology, Stanford University, Stanford, United States; [3]Department of Bioengineering, Stanford University, Stanford, United States

\*For correspondence:
proloy@stanford.edu

**Competing interest:** The authors declare that no competing interests exist.

**Abstract** Modern neurophysiological recordings are performed using multichannel sensor arrays that are able to record activity in an increasingly high number of channels numbering in the 100s to 1000s. Often, underlying lower-dimensional patterns of activity are responsible for the observed dynamics, but these representations are difficult to reliably identify using existing methods that attempt to summarize multivariate relationships in a post hoc manner from univariate analyses or using current blind source separation methods. While such methods can reveal appealing patterns of activity, determining the number of components to include, assessing their statistical significance, and interpreting them requires extensive manual intervention and subjective judgment in practice. These difficulties with component selection and interpretation occur in large part because these methods lack a generative model for the underlying spatio-temporal dynamics. Here, we describe a novel component analysis method anchored by a generative model where each source is described by a bio-physically inspired state-space representation. The parameters governing this representation readily capture the oscillatory temporal dynamics of the components, so we refer to it as oscillation component analysis. These parameters – the oscillatory properties, the component mixing weights at the sensors, and the number of oscillations – all are inferred in a data-driven fashion within a Bayesian framework employing an instance of the expectation maximization algorithm. We analyze high-dimensional electroencephalography and magnetoencephalography recordings from human studies to illustrate the potential utility of this method for neuroscience data.

## eLife assessment

This method article proposes a **valuable** oscillation component analysis (OCA) approach, in analogy to independent component analysis (ICA), in which source separation is achieved through biophysically inspired generative modeling of neural oscillations. The empirical evidence justifying the approach's advantage is **solid**. This work will be of interest to researchers in the fields of cognitive neuroscience, neural oscillation, and MEG/EEG.

## Introduction

Human neurophysiological recordings such as scalp electroencephalogram (EEG), magnetoencephalogram (MEG), stereoelectroencephalogram (SEEG), local field potentials, etc., consist of ~$10^2$ of sensors that record mixtures of predominantly cortical network oscillations (*Buzsáki, 2006*; *Buzsáki*

and Draguhn, 2004; Lopes da Silva, 2013; Wang, 2010; Helfrich and Knight, 2016). The network oscillations have distinct spatio-temporal signatures, based on the functional brain areas involved, their interconnections, and the electromagnetic mapping between the source currents and the sensors. However, the source-to-sensor mixing and the further superposition of measurement noise complicate the interpretation of the sensor-level data and its topography (Schaworonkow and Nikulin, 2022). Given the widespread and growing availability of high-density neural recording technologies (Stevenson and Kording, 2011), there is clearly a pressing need for analysis tools that can recover underlying dynamic components from highly multivariate data.

This problem fits within a larger class of blind source separation (BSS) problems for which there are a plethora of component analysis algorithms that attempt to extract underlying source activity as linear weighted combinations of the sensor activity. The decomposition weights are designed according to some predefined criterion on the extracted time series depending on the application. Independent component analysis (ICA) (Hyvärinen and Oja, 2000; Jung et al., 2000; Cardoso, 1999; Hyvärinen, 2013) has been particularly popular within neuroscience for a number of reasons. It requires no assumption of the original sources except for statistical independence, that is, no or minimal mutual information between the components (Bell and Sejnowski, 1995), and in principle requires little to no user intervention to run. However, a drawback of the method is that it assumes that the samples of each component time trace are independent and identically distributed, which is not generally true in physical or physiological applications. This leads to another major drawback of ICA: it relies on the cumulative histograms of sensor recordings and if these histograms are Gaussian, ICA is in principle unable to separate such sources (Brookes et al., 2011). This includes, for example, physiological signals such as EEG, MEG, sEEG, etc., that are at least approximately Gaussian distributed due to the central limit theorem (Muirhead, 1982) since these signals reflect the linear combination of many sources of activity. Finally, given the lack of assumptions on the structure of the underlying signals, there is no guarantee that ICA will extract relevant oscillatory components or any other highly structured dynamic pattern for that matter.

Given the prominence and ubiquity of oscillations in neurophysiological data, applications of component analysis methods in neuroscience have taken a different approach to emphasize oscillatory dynamics, employing canonical correlation analysis (Robinson et al., 2017), generalized eigenvalue decomposition (Parra and Sajda, 2003), joint decorrelation (de Cheveigné and Parra, 2014), or similar methods to identify a set of spatial weights to maximize the signal-to-noise ratio in the extracted component within a narrow band around a given frequency of interest (Nikulin et al., 2011; de Cheveigné and Arzounian, 2015; Cohen, 2017; Cohen, 2018). The extracted component then inherits the intrinsic oscillatory dynamics around that frequency without the need for a pre-designed narrow-band filter (Yeung et al., 2007). These techniques do acknowledge the inherent temporal dynamics of the oscillatory sources, but do so via nonparametric sample correlation or cross-spectrum matrix estimates that are sensitive to noise or artifacts and that require substantial amounts of data to achieve consistency.

Another problem common to all of these component decomposition methods is that they do not directly estimate the source to sensor mixing matrix. Instead, they estimate spatial filters that extract the independent components, which are not directly interpretable. The source to sensor mixing matrix can only be estimated by solving an inverse problem that requires knowledge of the noise covariance matrix (Haufe et al., 2014), which adds another complication and source of potential error. Finally, the properties of the extracted component time courses are not known a priori and must be assessed after the fact, typically by using nonparametric tests as well as visual inspection.

These difficulties with component selection and interpretation occur in large part because existing BSS methods lack a generative model for the underlying spatio-temporal dynamics. With a probabilistic generative model, it is possible to specify a soft constraint on the dynamic properties of the underlying components, while maintaining other desirable properties such as independence between components. Component selection can be handled automatically within a statistical framework under the model, and interpretation is straightforward in principle if the components can be described by a small number of parameters. Here, we propose a novel component analysis method that uses a clever state-space model (Wiener, 1966; Harvey, 1990; Harvey and Trimbur, 2003) to efficiently represent oscillatory dynamics in terms of latent analytic signals (Bracewell, 2000) consisting of both the real and imaginary components of the oscillation. The observed data are then represented as a superposition

of these latent oscillations, each weighted by a multichannel mixing matrix that describes the spatial signature of the oscillation. We estimate the parameters of the model and the mixing matrices using generalized expectation maximization (GEM) and employ empirical Bayes model selection to objectively determine the number of components. In these ways, we address the major shortcomings described above for many component analysis methods. We refer to our novel method as oscillation component analysis (OCA) akin to ICA. In what follows, we describe the model formulation in detail and demonstrate the performance of the method on simulated and experimental data sets, including high-density EEG during propofol anesthesia and sleep, as well as resting-state MEG from the Human Connectome Project.

## Theory
### State-space oscillator model

Oscillatory time series can be described using the following state-space representation (***Wiener, 1966***; ***Harvey, 1990***; ***Harvey and Trimbur, 2003***):

$$
\begin{aligned}
\mathbf{x}_n &= a\mathcal{R}(f)\mathbf{x}_{n-1} + v_n, \quad v_n \sim \mathcal{N}_2(0, \sigma^2 \mathbf{I}); \\
y_n^{(l)} &= \mathbf{c}_l^T \mathbf{x}_n + \epsilon_n, \quad\quad \epsilon_n \sim \mathcal{N}(0, R),
\end{aligned}
\tag{1}
$$

where the oscillation state, $\mathbf{x}_n = [x_{n,1}, x_{n,2}]^\top$, is a two-dimensional state vector. The stochastic difference equation summarizes oscillatory dynamics using random rotation in two-dimensional state space through a deterministic rotation matrix, $\mathcal{R}$, explicitly parameterized by the oscillation frequency, $f$, the sampling rate, $f_s$, and the damping factor, $a$, as

$$
\mathcal{R}(f) = \begin{bmatrix} \cos 2\pi f/f_s & -\sin 2\pi f/f_s \\ \sin 2\pi f/f_s & \cos 2\pi f/f_s \end{bmatrix}
\tag{2}
$$

and a stochastic driving noise, assumed to be stationary and Gaussian with variance $\sigma^2$. These elements of this state vector trace out two time series that maintains an approximate $\pi/2$ radian phase difference, and therefore are closely related to the real and imaginary parts of an analytic signal (***Bracewell, 2000***) in the complex plane (see Appendix 2, section 'Oscillation states and analytic signals'). The oscillation states are henceforth called analytic signal with minor abuse of notation. An arbitrary fixed projection (i.e., on the real line) of the state vector realizations generates the observed noisy oscillation time series at the sensor. Multiple oscillations can be readily incorporated in this state-space model by simply considering their linear combination. Recently, several investigators (***Matsuda and Komaki, 2017a***; ***Beck et al., 2018***; ***Beck et al., 2022***) utilized this state-space representation to extract underlying oscillatory time courses from single-channel EEG time traces.

### Generalization to multichannel data

In order to represent multichannel neural recordings, we employ this oscillatory state-space representation within a BSS model (***Parra, 1998***). In the proposed generative model, a sensor array with $L$ sensors records neurophysiological signals produced by superimposition of $M$ distinct oscillations supported by underlying brain networks or circuits, which we will refer to as *oscillation sources*.

$$
\begin{bmatrix} \mathbf{x}_t^{(1)} \\ \vdots \\ \mathbf{x}_t^{(M)} \end{bmatrix} = \begin{bmatrix} a^{(1)}\mathcal{R}\left(f^{(1)}\right) & \cdots & \mathbf{0} \\ \vdots & \ddots & \vdots \\ \mathbf{0} & \cdots & a^{(M)}\mathcal{R}\left(f^{(M)}\right) \end{bmatrix} \begin{bmatrix} \mathbf{x}_{t-1}^{(1)} \\ \vdots \\ \mathbf{x}_{t-1}^{(M)} \end{bmatrix} + \begin{bmatrix} v_t^{(1)} \\ \vdots \\ v_t^{(M)} \end{bmatrix}
$$

$$
\begin{bmatrix} y_t^{(1)} \\ \vdots \\ y_t^{(L)} \end{bmatrix} = \begin{bmatrix} \mathbf{c}_{1,1}^\top & \mathbf{c}_{1,2}^\top & \cdots & \mathbf{c}_{1,M}^\top \\ \mathbf{c}_{2,1}^\top & \mathbf{c}_{2,2}^\top & \cdots & \mathbf{c}_{2,M}^\top \\ \vdots & \vdots & \ddots & \vdots \\ \mathbf{c}_{L,1}^\top & \mathbf{c}_{L,2}^\top & \cdots & \mathbf{c}_{L,M}^\top \end{bmatrix} \begin{bmatrix} \mathbf{x}_t^{(1)} \\ \mathbf{x}_t^{(2)} \\ \vdots \\ \mathbf{x}_t^{(M)} \end{bmatrix} + \begin{bmatrix} \epsilon_t^{(1)} \\ \vdots \\ \epsilon_t^{(L)} \end{bmatrix}
\tag{3}
$$

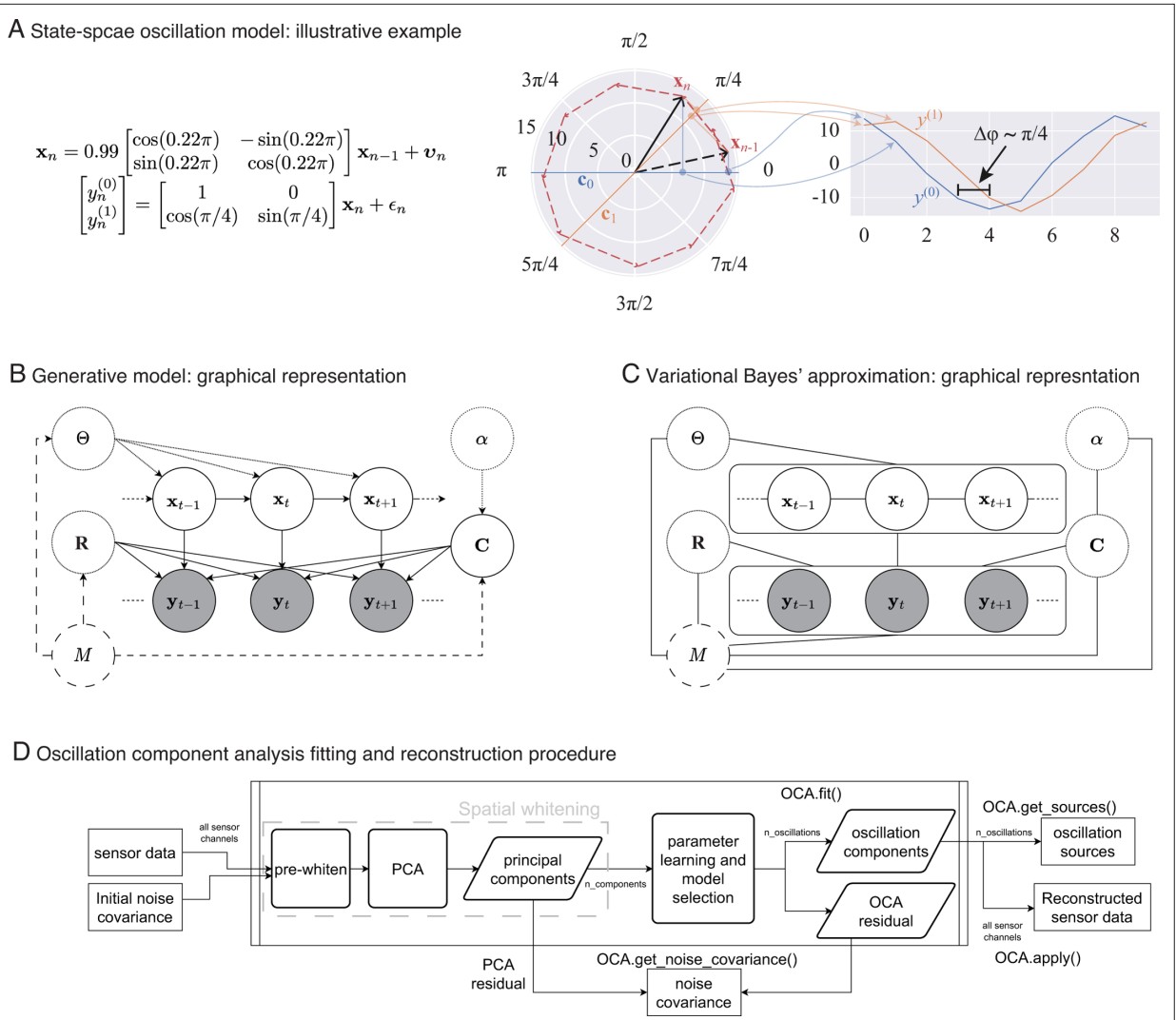

**Figure 1.** From state-space oscillator model to oscillation component decomposition. (**A**) An illustrative example of a multichannel state-space oscillation model: a single oscillation realized as an analytic signal $\mathbf{x}_n$ is measured as two different projections having radian phase difference. (**B**) Graphical representation of the probabilistic generative model describing the oscillations as dynamic processes that undergo mixing at the sensors and that are observed with additive Gaussian noise. (**C**) Graphical representation of the variational Bayes approximation that allows iterative closed-form inference. (**D**) Oscillation component analysis fitting and reconstruction pipeline for experimentally recorded neurophysiological data. The pipeline exposes a number of methods for ease of analysis, that is, for fitting the oscillation component analysis (OCA) hyperparameters `fit ()` method, which accepts the sensor recordings and an initial sensor noise covariance matrix as input, extracting the oscillation time courses `get_sources()`, reconstructing a multichannel signal from any arbitrary subset of oscillation components, `apply()`, getting a final noise covariance estimate from the residuals of OCA fitting, `get_noise_covariance()` method, etc.

Each oscillation source is governed by the above-mentioned state-space representation, resulting in a structured $2M$ dimensional state space. The structure of the underlying brain networks or circuits governs how each oscillation source is observed at the sensor array, via a $M \times 2$ spatial distribution matrix, $\mathbf{c}_{*,m} = [\mathbf{c}_{1,m}, \mathbf{c}_{2,m}, \cdots, \mathbf{c}_{L,m}]$. In other words, the $l$th electrode in the sensor array observes a specific projection of the underlying $m$th analytic oscillation given by $\mathbf{c}_{l,m} = [c_{l,m,1}, c_{l,m,2}]$, which encodes the amplitude and phase of the $m$th oscillation time course at that electrode. This probabilistic generative model for multichannel recordings is motivated by a potential biophysical mechanism of electromagnetic traveling wave generation in brain parenchyma (**Galinsky and Frank, 2020**; see Appendix 1, section 'Mechanistic origin').

## An illustration

In *Figure 1A*, middle panel, we depict an oscillation state as an analytic signal, $\mathbf{x}_n$ (black arrows), in 2D state space that is rotating around the origin according to the given state-space model of left panel: the red dashed line traces the oscillation states over time. The actual amount of rotation between every pair of consecutive timepoints is a random variable centered around $2\pi f/f_s = 0.22\pi$, with the spread determined by damping parameter, $a = 0.99$, and process noise covariance. The right panel shows two noisy measurements, reflecting two different projections: one on the real axis (blue traces), another on the line making $\pi/4$ radian angle to the real axis (orange traces). Because of this angle between the lines of projection, these two measurements maintain approximately $\pi/4$ radian phase difference throughout the time course of the oscillation.

We note that several earlier works *Matsuda and Komaki, 2017b*; *Quinn et al., 2021* have similar multivariate oscillator models, albeit from the perspective of a spatio-spectral eigendecomposition of the companion form of a multivariate autoregressive (MVAR) parameter matrix (*Neumaier and Schneider, 2001*). As described earlier, we employ this state-space form here in the context of a BSS problem.

Next, we briefly describe how one can infer the hidden oscillator states from observed multivariate time-series dataset given the oscillation state-space model parameters and potentially adjust the oscillation state-space model parameters to the dataset. We defer the mathematical derivations to Appendix 3 to maintain lucidity of the presentation.

## Learning algorithm

### Priors

We assume simple prior distributions on the measurement noise and sensor-level mixing coefficients to facilitate stable and unique recovery of the oscillation components. To obtain an $M$-oscillator analysis of $L$-channel data, we consider a Gaussian prior on the spatial mixing components, $\mathbf{c}_{l,m}$, with precision $\alpha$, and an inverse-Wishart prior on sensor noise covariance matrix, $\mathbf{R}$, with scale, $\boldsymbol{\Psi}$, and degrees of freedom, $\nu$:

$$
\begin{aligned}
p\left(\mathbf{C} \mid \alpha, M\right) &= \left(\frac{\alpha}{2\pi}\right)^{ML} \exp\left(-\frac{\alpha}{2} \sum_{l\geq 1, m\geq 1} \|\mathbf{c}_{l,m}\|^2\right) \\
p\left(\mathbf{R} \mid M\right) &= \gamma_{\nu,L} \left|\boldsymbol{\Psi}\right|^{\nu/2} \left|\mathbf{R}\right|^{-(\nu+L+1)/2} \exp-\frac{1}{2}\mathrm{Tr}\left\{\boldsymbol{\Psi}\mathbf{R}^{-1}\right\}
\end{aligned}
\tag{4}
$$

We treat the oscillation parameters, $(f^{(m)}, a^{(m)}, (\sigma^2)^{(m)}) := \boldsymbol{\theta}^{(m)}$, and distributional parameters of the assumed priors, $\alpha, \boldsymbol{\Psi}, \nu$, as hyperparameters. *Figure 1B* shows the probabilistic graphical model portraying oscillations as a dynamical system evolving in time, with the priors as parent nodes.

### Variational Bayes inference

Unfortunately, given the priors and the interplay between oscillation source time courses and sensor-level mixing patterns, exact computation of the log-likelihood, and thus the exact posterior, is intractable. We therefore employ variational Bayes (VB) inference (*Quinn and Šmídl, 2006*), a computationally efficient inference technique to obtain a closed-form approximation to the exact Bayes posterior. Originally introduced by *Hinton and van Camp, 1993*, VB inference simplifies the inference problem through a restrictive parameterization that reduces the search space for distributions, splitting up the problem into multiple partial optimization steps that are potentially easier to solve (*Attias, 1999*). The name comes from the *negative variational free energy*, also known as evidence lower bound (*Neal and Hinton, 1998*), used to assess the quality of the aforementioned approximation and as a surrogate for the log-likelihood (see Appendix 3, section 'Negative variational free energy').

In particular, given the number of oscillatory components $M$ and the corresponding hyperparameters $\boldsymbol{\theta} = [\boldsymbol{\theta}^{(1)}, \boldsymbol{\theta}^{(2)}, \cdots, \boldsymbol{\theta}^{(M)}]$ and $\alpha, \boldsymbol{\Psi}, \nu$ , we use the following VB decoupling (*Attias, 1999*) of the posterior distribution of mixing matrix $\mathbf{C}$, oscillation states $\mathbf{x}_t$, and noise covariance matrix $\mathbf{R}$:

$$
q\left(\mathbf{C}, \{\mathbf{x}_t\}, \mathbf{R} \mid \alpha, \boldsymbol{\theta}, M\right) = q\left(\mathbf{C} \mid M\right) q\left(\{\mathbf{x}_t\} \mid M\right) q\left(\mathbf{R} \mid M\right)
\tag{5}
$$

This particular choice allows the approximate posteriors of these quantities to be represented in closed form by multivariate Gaussian, multivariate Gaussian, and inverse-Wishart distributions,

respectively (see Appendix 3, section 'Variational Bayes inference'). *Figure 1C* shows the graphical representation of posterior distribution after the VB decoupling. This essentially allows us to perform an iterative posterior inference procedure, where we cyclically update the posteriors $q(\mathbf{x}_t \mid M)$, $q(\mathbf{C} \mid M)$, and $q(\mathbf{R} \mid M)$ using the latest sufficient statistics from other two distributions (see Appendix 3, section 'Variational Bayes inference' for more details).

## Generalized EM to update hyperparameters

Since the parameters of the state-space model, and of the assumed prior distributions, are not known a priori, we then obtain their point estimates using an instance of the GEM algorithm, which utilizes the aforementioned approximate inference in the E-step (*Dempster et al., 1977*; *Neal and Hinton, 1998*). We start the learning algorithm by initializing $\theta$, $\overline{\Lambda}$, $\overline{C}$, $\alpha$ to appropriate values (see Appendix 3, section 'Initialization of parameters and hyperparameters' for details). We then cyclically update the posteriors $q(\mathbf{x}_t \mid M)$, $q(\mathbf{C} \mid M)$, and $q(\mathbf{R} \mid M)$ until these iterations stop changing the negative variational free energy. At the end, we update the hyperparameters $\theta$, $\alpha$ from the current inference and proceed to the next inference with the updated hyperparameters. These update rules form an outer update loop that refines the hyperparameters, within which operates an inner update step that iteratively improves the approximate posterior distribution of the model parameters. We defer the update rules to Appendix 3, section 'Generalized EM algorithm'. The algorithm terminates when the hyperparameter adjustments no longer increase the free energy of the model, that is, we resort to parametric empirical Bayes estimation (see Appendix 3, section 'Empirical Bayes inference and model selection).

## Selecting optimal number of oscillations

The learning algorithm assumes that the number of oscillation sources, $M$, is known a priori, which is rarely the case for experimentally recorded data. In theory, our probabilistic treatment could assign a log-likelihood for a given dataset to every model structure, that is, the number of oscillation sources, that can be used as a goodness-of-fit score. Unfortunately, the log-likelihood is intractable to evaluate; however, the negative variational free energy can be used as a surrogate.

We formalize this idea by considering that the model structure, $M$, is drawn from a discrete uniform distribution over a finite contiguous set, $(1, M_{max})$:

$$p(M) = 1/M_{max}, \quad 1 \leq M \leq M_{max}, \tag{6}$$

with $M_{max}$ being the maximal number of oscillation sources. This allows us to compute the posterior on the model structure, $q(M)$, within the same variational approximation framework. Using the negative variational free energy expression of the oscillator model, it is easy to show that $q(M)$ is proportional to the exponential of the negative variational free energy of the $M$-oscillator model (see Appendix 3, section 'Model structure posterior'). Once the model posteriors are obtained, one can choose to simply select the model with maximum negative variational free energy (empirical Bayes model selection, see Appendix 3, section 'Empirical Bayes inference and model selection'), or locate the *knee* or *jump* point that exhibits a significant local change in $q(M)$ (for such an example, see the DiffBic function; *Zhao et al., 2008*).

## Oscillation component analysis

A preliminary version of this algorithm has been presented previously (*Das and Purdon, 2022*). In order to analyze experimental data, we devised the a standardized pipeline as demonstrated in *Figure 1D* closely following the MNE `ICA` pipeline (*Gramfort et al., 2014*). We refer to this pipeline as `OCA`. The pipeline, implemented as Python class `OCA`, accepts the sensor recordings either as continuous raw data or a set of discrete epochs, each with an initial sensor noise covariance matrix. The hyperparameters for the measurement noise covariance prior are derived from the supplied initial sensor noise covariance matrix and are not updated in this pipeline. The sensor data is standardized (i.e., pre-whitened) against the sensor noise covariance: this step also applies all active signal-space projectors to the data if present such as the average reference (*Uusitalo and Ilmoniemi, 1997*). The pre-whitened data is then decomposed using PCA. The first `n_components` are then passed to the parameter learning iterations for hyperparameters $\theta$, $\alpha$, followed by the joint inference of the mixing matrix, oscillation time courses, and residual noise covariance. The optimal number of oscillations is

then selected as described earlier. The final noise covariance is computed from the remaining PCA components that are not supplied to oscillation fitting and the OCA residuals. Once the oscillation hyperparameters and the mixing matrix have been estimated, the oscillation time courses can be estimated from any continuous recordings or discrete epochs. Finally, the pipeline can also reconstruct a multichannel signal from any arbitrary subset of oscillation components. We emphasize here that the main reason for using PCA here is to perform rotation of the data, so that (1) rank-deficient data (i.e., average referenced EEG data) can be handled conveniently by throwing out the zero-variance component. (2) The variance the multichannel data can be uniformly distributed in the remaining PCs. For these reasons, we retain components explaining 99.9% variance of the multichannel recordings, that is, almost all components with nonzero variance. The PCA step essentially performs rotation of the data in the spatial dimension, leaving the temporal relationships untouched.

## Results

Finally, we apply OCA on a simulation example and several experimentally recorded datasets. We demonstrate the versatility of OCA analysis using two EEG datasets, one under propofol-induced anesthesia, another during sleep, and one resting-state MEG dataset. The simulation example shows the utility of such time-domain analysis over frequency-domain analysis in the context of multichannel recordings. In the real data applications, we showcase the implications of OCA as a mechanistically grounded tool, suggesting various downstream analysis involving the oscillation time courses and their sensor distributions.

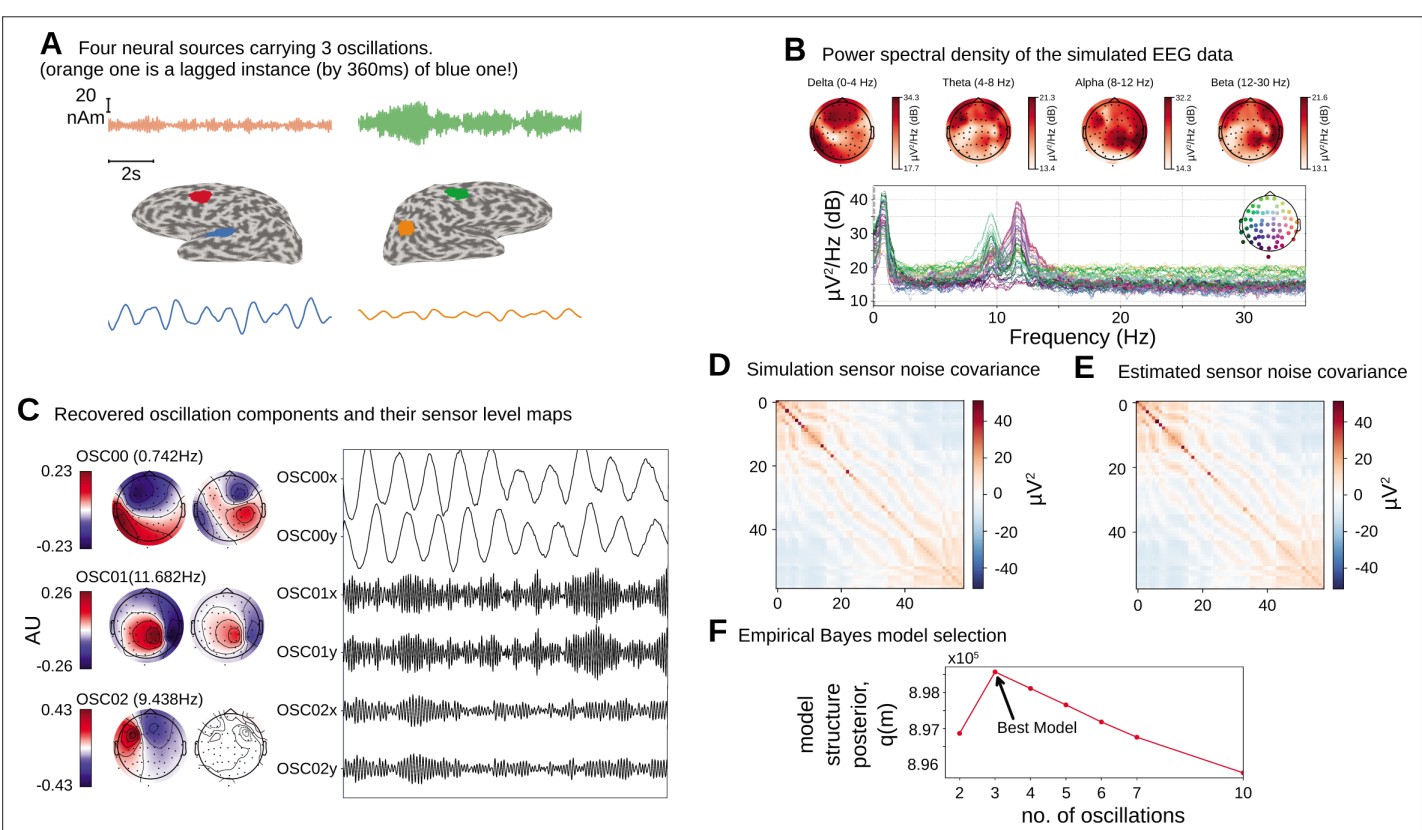

**Figure 2.** Simulation study. (**A**).Four neural sources carrying three oscillations, where the orange time course is a lagged instance of blue time course, red and green time courses are independent. (**B**) Power spectral density of the simulated electroencephalogram (EEG) recording and power distribution over the EEG sensors in different frequency bands. (**C**) Recovered oscillation components and their sensor-level maps. (**D**) Sensor noise covariance for the simulation. (**E**) Estimated sensor noise covariance matrix. (**F**) Model structure selection via model structure posterior $q(m)$.

## Simulation example

We first apply the proposed oscillation decomposition method on a synthetic dataset to illustrate how our time-domain approach differs from traditional frequency-domain approach. We generated a synthetic EEG dataset using a 64-channel montage, forward model and noise covariance matrix from a sample dataset distributed with the MNE software package (*Gramfort et al., 2014*). To select the active current sources, four regions with 5 mm$^2$ area were centered on chosen labels in the DKT atlas (*Desikan et al., 2006*): 'transversetemporal-lh', 'precentral-rh', 'inferiorparietal-rh', and 'caudalmiddlefrontal-lh'. Three time courses were separately generated at a sampling frequency of 100 Hz according to univariate AR(2) dynamics tuned to generate stable oscillations at 1.6 Hz (slow/delta), 10 Hz (alpha), and 12 Hz (alpha) frequencies. The first two regions were simulated with the same slow/delta time course such that the activity in the 'precentral-rh' area (*Figure 2A*, orange) lags that of 'transversetemporal-lh' area (*Figure 2A*, blue) by 10 ms. The last two regions, 'inferiorparietal-rh' and 'caudalmiddlefrontal-lh', were simulated with two independent alpha time courses. We projected this source activity to the EEG sensors via a lead-field matrix and add spatially colored noise generated with the covariance structure in *Figure 2D* to the sensor data. Two 20 s epochs were chosen for OCA and model selection was performed within the range of (2, 3, 4, 5, 6, 8, 10) oscillations. *Figure 2B* presents the two most common frequency-domain visualizations of multichannel data: the top panel shows the power distribution over the scalp within different canonical EEG frequency bands, while the channel-wise power spectral density (PSD) is shown in the bottom panel. *Figure 2C* demonstrates how the decomposition obtained by OCA provides an accurate delineation of different oscillation time courses and effective characterization of their spatial distributions in the sensor-level mixing maps. Unlike the frequency-domain techniques, the residual time series that follows OCA extraction provides the temporally unstructured part of the multichannel recording. The estimated covariance matrix from these residuals closely resembles the original sensor noise covariance matrix that was used to corrupt the recordings (see *Figure 2E*). The covariance matrix estimate is strikingly close to the true covariance matrix (i.e., compare with *Figure 2D*). OCA provides a measure, $q(M)$, for model structure selection that objectively identifies the existence of three oscillation time courses with different time dynamics (see *Figure 2F*).

## How does OCA compare to conventional approaches?

We performed additional analyses to compare OCA with Fourier-based frequency domain method and ICA in a different realization of a similar synthetic EEG dataset (see *Figure 3A*). The frequency-domain approach applied here is based on the multitaper method: given a bandwidth parameter of 2 Hz, power spectrum density is computed for individual channels. The oscillatory activity can be identified visually as the peaks in the power spectrum plot in *Figure 3B*, top panel, and its scalp distribution can be obtained by averaging the power within given frequency bands around the peaks in each channel (as shown in *Figure 3B*, bottom panels). However, these visualizations do very little to identify underlying oscillatory sources and pose additional questions. For example, the left topographical plot hints to two possible sources, but provides no evidence if they are separated spatially, temporally or both. Similarly, the right two topographical plots are almost identical and show no separability between these the oscillatory sources that generate two distinct peaks in spectrum. Regarding ICA, we use the 'extended infomax' (*Lee et al., 1999*) implementation provided by MNE-python 1.2 (*Gramfort et al., 2014*) to see if ICA can distinguish the three statistically independent sources in these data. The leading four ICA components are shown in *Figure 3C*. The ICA-identified components mix the underlying independent signals, which is not surprising since all components follow a Gaussian distribution. Meanwhile, OCA is able to recover three distinct oscillatory components consistent with the ground truth. OCA is able to do so because it explicitly models temporal dynamics of the components.

We repeated similar analyses using real human EEG recording and found a similar result where ICA produced components that mixed slow and alpha band signals, whereas OCA identified distinct oscillatory components (see Appendix 4, section 'Comparison of OCA to traditional approaches in experimental EEG data').

## EEG recording during propofol-induced unconsciousness

Next, we demonstrate utility of OCA on experimental data using EEG recordings from a healthy volunteer undergoing propofol-induced unconsciousness, previously described in *Purdon et al.,*

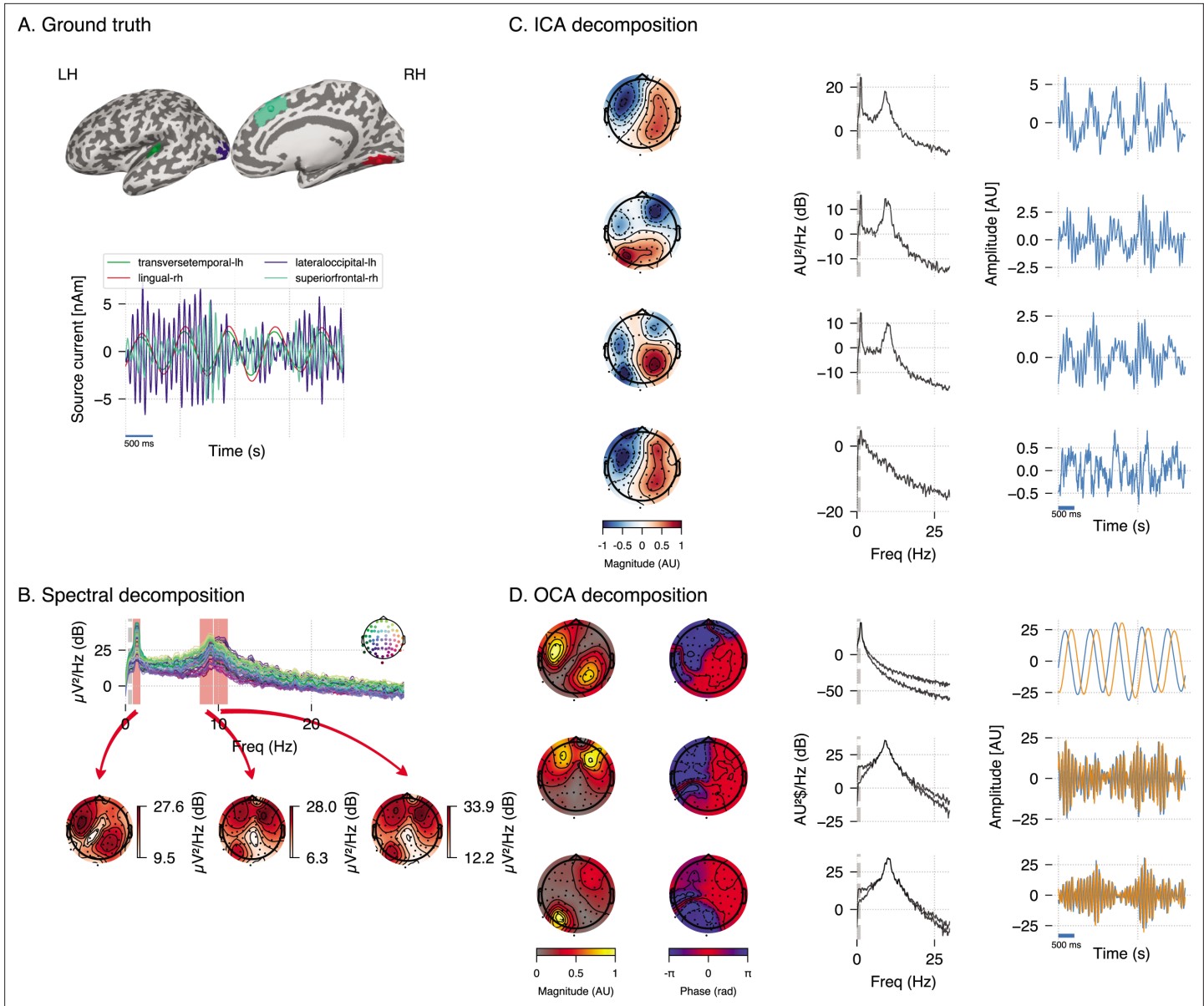

**Figure 3.** Simulation study (extended). (**A**) Four neural sources carrying three oscillations, similar to *Figure 2*. (**B**) Power spectral density of the simulated electroencephalogram (EEG) recording and power distribution over the EEG sensors in the frequency bands (red overlay) around visually identifiable peaks. (**C**) Recovered independent component analysis (ICA) components (left, middle, and right columns show topographic maps, power spectrum density, and time courses, respectively). (**D**) Recovered oscillation component analysis (OCA) components (the topographic maps show the magnitude [left] and phase [right], while line plots show power spectrum density [left] and time courses [right], respectively).

2013. For induction of unconsciousness, the volunteer underwent a computer-controlled infusion of propofol to achieve monotonically increasing levels of effect-site concentration in steps of $1\ \mu g\ \text{mL}^{-1}$. Each target effect-site concentration level was maintained for 14 min. The EEG was recorded using a 64-channel BrainVision MRI Plus system (Brain Products) with a sampling rate of 5000 Hz, bandwidth 0.016–1000 Hz. The volunteers were instructed to close their eyes throughout the study. Here, we investigated EEG epochs during maintenance of effect-site concentrations of $0\ \mu g\ \text{mL}^{-1}$ (i.e., baseline, eyes closed), $2\ \mu g\ \text{mL}^{-1}$, and $4\ \mu g\ \text{mL}^{-1}$ for OCA. We selected 10 clean 3.5 s epochs corresponding to those target effect-site concentration (see 'Materials and methods').

We fitted OCA models with 20, 25, 30, 35, 45, 50, 55, 60 oscillation components and selected the model with highest negative variational free energy as the best OCA model. Following these empirical Bayes criteria, we selected OCA models with 30, 30, 50 components for these three conditions,

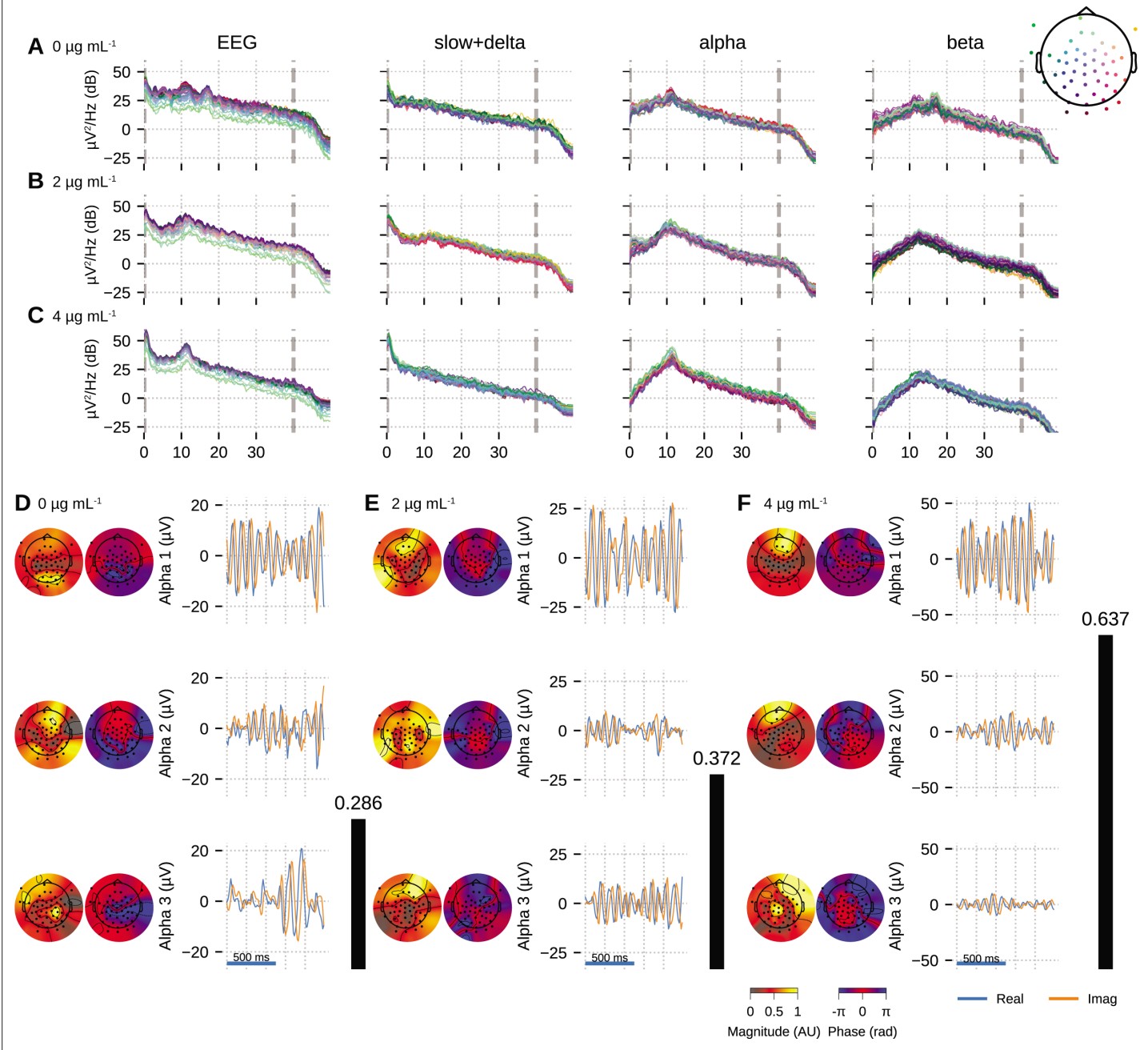

**Figure 4.** Oscillation component analysis (OCA) of the electroencephalogram (EEG) from a healthy volunteer undergoing propofol-induced unconsciousness. Conditions of target effect-site concentration of (**A**, **D**) 0 (i.e., baseline), (**B**, **E**) 2 µg mL$^{-1}$, and (**C**, **F**) 4 µg mL$^{-1}$ are analyzed. Panels (**A–C**) show the power spectral densities (PSDs) of reconstructed EEG activity within each canonical band. Panels (**D–F**) show the three dominant (in terms of sensor wide power) alpha component: the topographic maps show the magnitude (left) and phase (right) distribution of sensor-level mixing, the time courses are 1 s representative example of the extracted oscillations from the selected epochs. The black bars on the right display the coherency measure within alpha band. OCA correctly identifies that the spatial mixing sensor maps of the alpha waves (8–12 Hz) are oriented posteriorly at baseline, but gradually become frontally dominant under propofol. The sensor weights are scaled to have maximum value 1. So, the units of time series traces can be considered to be in µV.

respectively. When the center frequencies were grouped within the canonical frequency bands (**Buzsáki, 2006**), we found 17, 17, 29 slow/delta (0.1–4 Hz) oscillation components and 17, 9, 15 alpha (8–13 Hz) oscillation components for the three conditions, respectively. **Figure 4A–C** shows PSDs of reconstructed EEG activity within each band, defined as the cumulative projection of these grouped oscillation components to the sensors. Since each of these oscillations have different sensor

distributions, the increasing number of oscillation components can be associated with fragmentation of neuronal networks that support these oscillations.

We also quantified the coherency of the reconstructed EEG activity within these commonly used frequency bands as the ratio of the power of the strongest component to the total power of the oscillation components within the bands. For the data analyzed here, slow + delta band coherency decreases gradually while the alpha band coherency increases with the increasing propofol effect-site concentration. To investigate this observation further, we visualized three dominant alpha components from each conditions in order of their strengths in *Figure 4D–F*. In each column, the left sub-panels consist of two topographic plots: the left one showing the distributions of the strength (scaled between 0 to 1) and the right one showing the relative phase of the estimated analytic oscillation mixing, $\mathbf{c}_{i,j}$. The right sub-panels display examples of the extracted analytic oscillations, that is, estimated real and imaginary components of 1 s long oscillation segments. The rightmost black bars show the coherency measure within alpha band. Clearly, during the $4\ \mu g\ mL^{-1}$ effect-site concentration, the amplitude of the leading alpha oscillation component is much bigger than the others, which aligns with our observation of the coherency measure. This finding previously reported analyses showing a rise of slow + delta oscillation power, rapid fragmentation of a slow + delta oscillation network, and emergence of global alpha network during propofol-induced anesthesia (*Lewis et al., 2012*; *Murphy et al., 2011*). Further, as the propofol effect-site concentration increases, the temporal dynamics increasingly resemble a pure sinusoidal signal, as evident from the PSD of the alpha band reconstruction that becomes more concentrated around the center frequency. Lastly, the topographic plots in *Figure 4D–F* illustrate how alpha activity shifts from posterior electrodes to frontal electrode with increasing propofol concentration (*Cimenser et al., 2011*; *Purdon et al., 2013*).

## EEG recordings under different sleep stages

The HD-EEG data shown in *Figure 5* is from a young adult subject (age 28, female) undergoing a sleep study, recorded using the ANT Neuro eego mylab system. After preprocessing (see 'Materials and methods'), we visually selected 10 clean 5 s segments of data that exhibited strong alpha oscillations, each from relaxed wakeful (eyes closed) and rapid eye movement (REM) sleep to apply OCA.

We fitted OCA models with 20, 25, 30, 35, 45, 50, 55, 60 oscillation components and selected the model structure with the highest negative variational free energy as the optimum OCA model. The empirical Bayes criterion selected 55 and 50 OCA components for relaxed wakeful (eyes closed) and REM sleep, respectively. *Figure 5* follows a format similar to *Figure 4*: panels A nd B show the power spectrum density of the recorded EEG and the reconstructed sensor data from the OCA components grouped in slow, theta and alpha bands; while panels C and D visualize three of the extracted oscillation components within the alpha band, ordered according to their strengths. The topographical maps show the scalp distribution of these components, while the blue and red traces show 1-s-long oscillation time-series pair corresponding to each component.

Activity within the alpha band (8–13 Hz) could be summarized by a few oscillation components: for example, only seven and three OCA components had their center frequency within alpha band during relaxed wakeful (eyes closed) condition and REM sleep, respectively. These alpha oscillation components during relaxed wakeful (eyes closed) condition and REM sleep appeared to be distinct in terms of their temporal dynamics: relaxed wakeful alpha components are more regular (i.e., closer to sinusoidal activity) than REM alpha. But REM alpha coherency (0.4307) is moderately higher than that of relaxed wakeful alpha (0.3195). The spatial distribution of the REM alpha component appeared to be confined primarily within the posterior channels while the awake alpha is more distributed. To quantify the degree of similarity or difference between the spatial distributions, we calculated the principal angle between the spatial mixing maps of the awake eyes closed and REM alpha components. The *principal angle* measures the similarity between two subspaces, where an angle of 0° indicates that one subspace is a subset of the other and an angle of 90° indicates that at least one vector in a subspace is orthogonal to the other (*Bjorck and Golub, 1973*). The spatial mixing maps for awake eyes closed alpha and REM sleep had a principal angle of 44.31°, suggesting that the subspaces, and thus the underlying cortical generators, were substantially different.

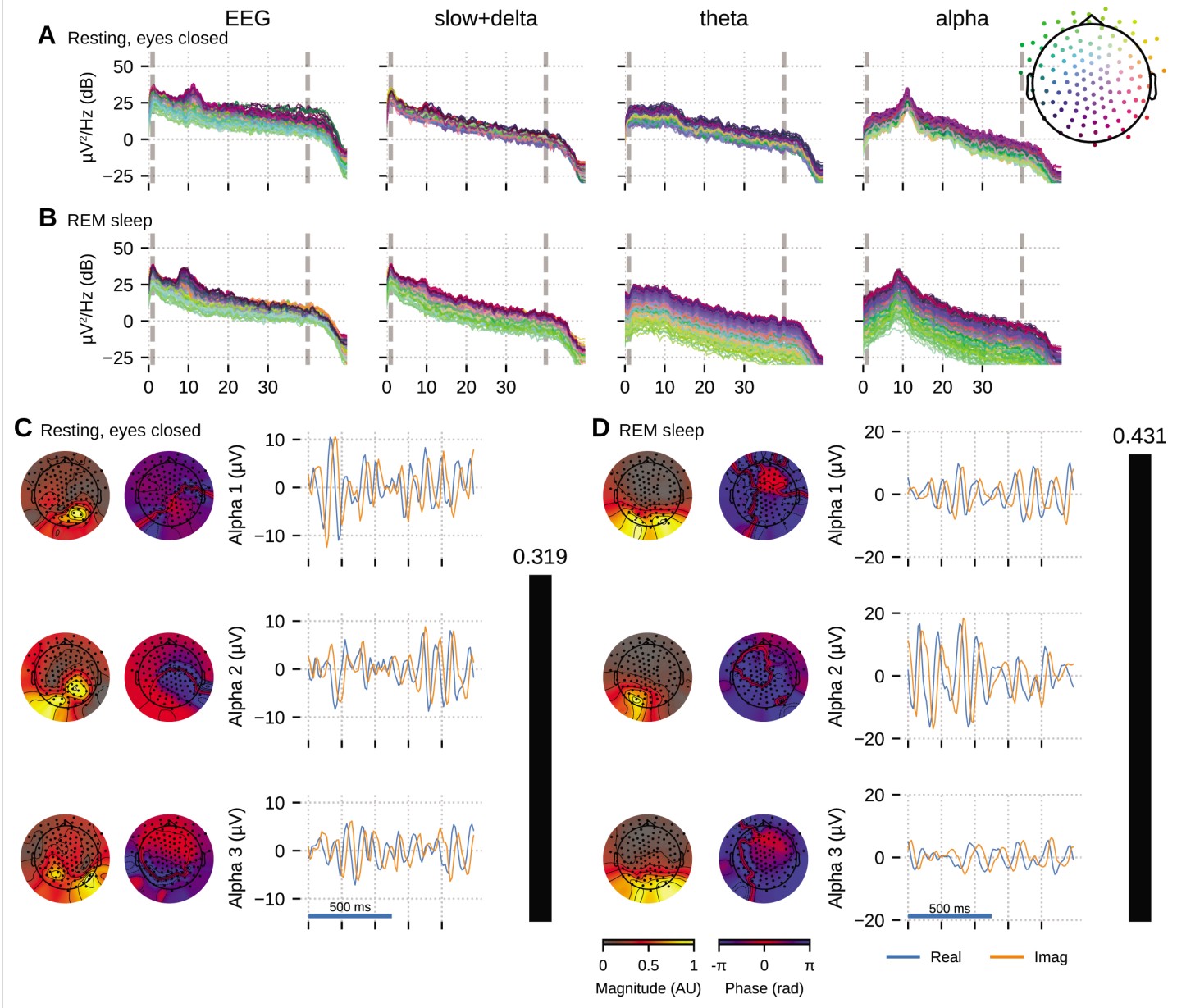

**Figure 5.** Oscillation component analysis (OCA) of the electroencephalogram (EEG) from a healthy young volunteer to compare between (**A**, **C**) wakeful resting state and (**B**, **D**) rapid eye movement (REM) sleep. Panels (**A**, **B**) show the power spectral densities (PSDs) of reconstructed EEG activity within each canonical band. Panels (**C, D**) show the three dominant (in terms of sensor-wide power) alpha component: the topographic maps show the magnitude (left) and phase (right) distribution of sensor-level mixing, the time courses are 1 s representative example of the extracted oscillations from the selected epochs. The rightmost black bars display the coherency measure within alpha band. The contrasting topographic distribution of the alpha components (8–12 Hz), the shape of the oscillation power spectrum and alpha coherence hints at a distinct generating mechanism for alpha waves during stage 2 REM sleep compared to awake eyes closed alpha wave. The sensor weights are scaled to have a maximum value 1 so that the units of time series traces are in µV.

## Resting-state MEG recording

Finally, we demonstrate the utility of OCA on a MEG dataset from the Human Connectome Project. The MEG data used here is the resting-state recording from subject 104 012_MEG, session 3.

We fit OCA on 34 clean 5s epochs, and from among the OCA with 20, 25, 30, 35, 40, 45, 50, 55 components, empirical Bayes criterion selected the OCA with 30 components. Out of these, 13 were within the slow + delta band and 10 were alpha oscillations. For this example, we sought to highlight a different downstream analysis of the extracted oscillation components. We chose the three leading

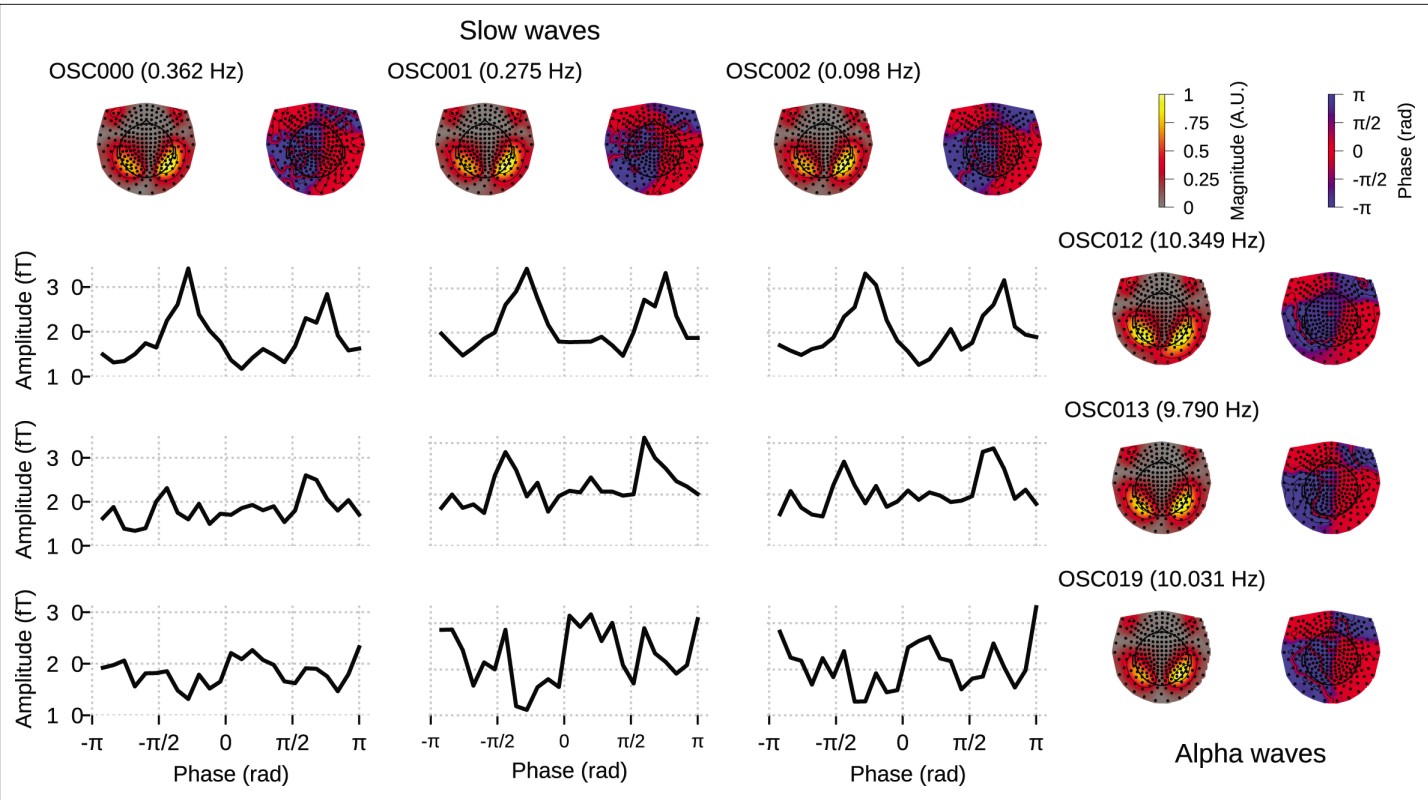

**Figure 6.** Cross–frequency phase–amplitude coupling in oscillation component analysis (OCA) components extracted from resting-state magnetoencephalogram (MEG) recording. The black traces show the conditional mean of a selected alpha component (8–12 Hz) amplitude given another selected slow/delta component (0–4 Hz) phase. The three slow oscillations and three alpha oscillations that explained the highest variance were selected for demonstration purposes. The topographic maps show the magnitude (left) and phase (right) distribution of sensor-level mixing of the selected components.

slow oscillation components and the three leading alpha oscillation components from the set of identified oscillation components.

For each oscillation, OCA extracts a pair of time traces, that is, one *real* time trace and one *imaginary* time trace (see Appendix 2, section 'Oscillation states and analytic signals'). These 'real' and 'imaginary' indices can be utilized to compute the instantaneous phase and instantaneous amplitude of individual oscillation components at every timepoint, without having to rely on Hilbert transform (*Wodeyar et al., 2021*).

$$A_t^{(*)} = \sqrt{x_{t,1}^{(*)^2} + x_{t,2}^{(*)^2}}, \qquad P_t^{(*)} = \arctan\left(\frac{x_{2,t}^{(*)}}{x_{1,t}^{(*)}}\right)$$

Further, we pick any one of the slow oscillation components and one of the alpha oscillation components and consider the ordered pair of slow component phase and alpha component amplitude at a single timepoint as a sample drawn from the joint distribution of the 'phase–amplitude' of these two components. These samples then can be used to quantify cross frequency phase–amplitude coupling between these components via either nonparametric (*Cohen, 2008*; *Cheng et al., 2018*; *Martínez-Cancino et al., 2019*) or parametric modeling (*Soulat et al., 2022*; *Perley and Coleman, 2022*).

*Figure 6* demonstrates a possible way of investigating the coupling between the slow oscillation component phase and alpha oscillation component amplitude: the conditional mean of the alpha amplitude given slow/delta phase is higher at some specific phases between all three slow waves and `OSC012`, but overall flat for other alpha oscillation components. This example illustrates how the OCA algorithm can be used to identify specific oscillatory components that show phase–amplitude coupling.

## Discussion

OCA is a novel approach to the multichannel component decomposition problem that can identify a number of independent spatio-temporal components, but does so in the context of underlying temporal dynamics specified by a state-space model. In the state-space representation used here, the dynamics of an elemental oscillation are parameterized by a central frequency, $f$, a damping parameter, $a$, and the second-order statistics of stochastic driving noise, $\sigma^2$. Meanwhile, an associated spatial mixing pattern at the sensor level quantifies the contribution of the elemental oscillation to each observed channel.

The OCA learning algorithm uses an instance of the GEM algorithm to iteratively *match* the parameters of the oscillation state-space parameters to the second-order statistics of the M/EEG data. The goodness-of-fit for the data-driven matching procedure is defined within a Bayesian framework, as is the inference for the oscillation time courses, their sensor-level mixing patterns, and the measurement noise covariance matrix. This same goodness-of-fit metric is further used to determine the number of oscillations present in the multichannel data via empirical Bayes model selection. Once the number of oscillations, oscillation state-space parameters, their sensor-level mixing patterns, and the measurement noise covariance matrix are estimated, the elemental oscillatory activities can be extracted as a pair of time courses, that is, the 'real' and 'imaginary' parts of the analytical signal, from any given observation.

The oscillator state-space parameters discovered in OCA are akin to the ad hoc parameters, for example, peak frequency, full-width-half-maxima bandwidth, and peak oscillation power, etc., encountered in Fourier/wavelet-based frequency-domain nonparametric time-series analysis (*Donoghue et al., 2020*). However, OCA circumvents a major drawback of frequency-domain methods when searching for neural oscillations from multichannel recordings. Generally, the source-to-sensor mixing complicates the interpretation of the topographies generated by signal processing tools that treat each channel individually (*Schaworonkow and Nikulin, 2022*). In a nonparametric frequency-domain approach, the only way to discover this spatially correlated structure across sensors is to perform eigenvalue analysis on the cross-spectral density on a frequency-by-frequency basis, also known as global coherence analysis (*Cimenser et al., 2011*; *Weiner et al., 2023*; *Mitra and Bokil, 2007*). If the eigenvalue distribution exhibits sufficient skewness, the associated frequency is deemed coherent and the leading eigenvectors are identified as the 'principal' sensor networks at that frequency; in other words, a proxy for a strong oscillatory component. As the number of sensors increases, this analysis becomes increasingly intractable: in a sensor array with $\sim 10^2$ sensors, the cross-spectral density matrix has $\sim 10^4$ entries. Alternatively, the state-space representation used here in OCA provides a convenient and compact way to perform multichannel time-domain modeling and analysis (*Nise, 2011*). The OCA modeling approach decouples the oscillatory dynamics and the spatial mixing pattern effectively by estimating one set of parameters for each discovered oscillation and the associated spatial mixing matrix, thereby avoiding the need for cross-spectral density matrices and frequency parameters for individual channels altogether (*Gunasekaran et al., 2023*). Another major advantage of OCA over global coherence analysis is that OCA automatically identifies the dominant *coherent* oscillations across the channels based on its probabilistic generative model and Bayesian learning approach.

The iterative parameter estimation procedure of OCA is clearly more computationally burdensome compared to conventional frequency-domain nonparametric methods. However, once the OCA parameters are estimated, OCA can decompose any given multichannel recording segments into the oscillation time-series pairs. In that sense, OCA parameter learning can be viewed as iterative estimation of a pair of optimal spatio–temporal filters (*Anderson and Moore, 2005*) for each oscillation, parameterized by the estimated oscillation state-space parameters and spatial maps. These optimal spatio-temporal filters are then applied to the multichannel data to extract the oscillation time courses. As a result, the extracted narrowband activity is endogenous to the data, rather than being imposed by an arbitrary narrowband filter (*Yeung et al., 2007*).

This behavior of OCA is similar to oscillatory component extraction methods based on data-driven spatial filtering, where the extracted component inherits the intrinsic oscillatory dynamics around the specified frequency (*Nikulin et al., 2011*; *de Cheveigné and Arzounian, 2015*; *Cohen, 2017*; *de Cheveigné and Arzounian, 2015*). In fact, OCA employs the same philosophy as spatio-temporal source separation (*Cohen, 2018*), where the extracted narrow-band signal and its time-shifted versions are again projected to a temporal coordinate space to further enhance the power within the given

frequency band. However, spatio-temporal source separation establishes the temporal constraint via nonparametric sample correlation estimates that are sensitive to noise or artifacts and that require substantial amounts of data for estimation. Another important distinction is that spatio-temporal source separation determines the sensor weights first and then obtains the temporal filter kernel from the projection of the multichannel data. In contrast, OCA updates the sensor weights and the parameters for the temporal filter iteratively within the Bayesian formulation of the state-space framework, thus jointly estimating the spatial and temporal components simultaneously from the data.

The class of ICA-based methods, on the other hand, assumes that the component time courses share no mutual information, that is, are statistically independent. Without any explicit assumption on the temporal dynamics of the generative process, the properties of the identified components may be difficult to interpret and may also be unreliable. In fact, the identified ICA components typically require subjective visual inspection by experts for their possible interpretation. A number of investigators have recognized these limitations with ICA, in particular the assumption of temporal independence, and have proposed generalizations of the ICA framework to incorporate auto-regressive modeling of source dynamics (*Pearlmutter and Parra, 1996*; *Parra, 1998*). Despite demonstrating improved source separation performance in naturalistic signals like music (*Pearlmutter and Parra, 1996*), adoption of these methods for the analysis of neurophysiological data has been slow. This may be due in part to the lack of interpretability of the higher order auto-regressive model structures. Alternatively, *Brookes et al., 2011* use a combination of Hilbert envelope computation within predefined frequency bands and temporal downsampling to identify meaningful temporally independent time signals, but stop short of trying to disambiguate different oscillatory sources. Thus, they only assess the connectivity pattern within predefined bands, that is, how different areas of the brain are harmonized through modulation of the oscillations or vice versa inside those predefined bands. The spatial maps recovered from anatomically projected resting-state MEG data by this method resemble spatial patterns of fMRI resting-state networks. OCA is close to the first variant of ICA, but describes the identified component sources using a 2D state-space vector that efficiently represents oscillatory dynamics (*Matsuda and Komaki, 2017a*) in a manner that is easy to interpret. *Matsuda and Komaki, 2017b* described a similar state-space model for multivariate time series, but did not apply the model to neurophysiological data, perhaps due to its high dimensionality. The OCA algorithm differs by way of its learning and model selection algorithms, which allow it to select the number of components to extract in a statistically principled and data-driven manner. Overall, OCA delivers the best of the previously mentioned eigen-analysis-based source separation methods and ICAs: a decomposition method that can identify independent components, but where each component represents an underlying oscillatory dynamical system. The generative link between each component and the underlying dynamical system makes interpretation of the components straightforward. Here, we focus specifically on analyzing neurophysiological data that are exemplified by highly structured oscillatory dynamics. But the methods we describe apply to more general, arbitrary dynamics that can be approximated by linear state-space models, and can be equally useful so long as the state-space model is interpretable.

There is another important distinction between OCA and the aforementioned BSS techniques. These methods directly estimate the sensor weights for a weighted linear combination of sensor recordings that produce the individual temporal components (*Haufe et al., 2014*). In other words, these BSS methods provide a backward decoding algorithm. The linear weights are not directly interpretable since they do not specify how the individual temporal components map to the observed data. That uniquely interpretable information is provided by a forward mapping which must be estimated in a separate step. When the number of linear components matches with the number of sensors, this forward mapping can be calculated via simple matrix inversion. However, when there is a reduced set of components the forward mapping can only be obtained by solving an inverse problem that requires knowledge of the source and sensor sample covariance matrix. In contrast, the OCA model involves only the (forward) spatial distribution matrix and our algorithm directly estimates it. In fact, the backward extraction model of OCA involves nonzero weights for time-shifted signals and is never explicitly computed. In that sense, OCA avoids an extraneous transformation step that could inadvertently introduce errors to the spatial mixing pattern estimates.

Another popular time-domain approach for oscillatory signal discovery is the multichannel extension of empirical mode decomposition. Empirical mode decomposition models the time-series data as linear combination of intrinsic oscillations called intrinsic mode functions (IMFs). Identification of

IMFs depends critically on finding the local mean of the local extrema, which is not well-defined in the context of multichannel recordings. *Rehman and Mandic, 2010* chose to take real-valued projections of the *n*-channel data, with the projections taken along direction vectors uniformly sampled on a unit spherical surface in an *n*-dimensional coordinate space. The local extrema of each of the projected signals are extrapolated to obtain a set of multivariate signal envelopes. Clearly, this random sampling in a high-dimensional space is computationally demanding, and the fully nonparametric formulation requires a substantial amount of data, making the procedure sensitive to noise. On the contrary, OCA employs a parametric model that represents oscillatory dynamics in a manner that is statistically efficient and resilient to noise.

## Conclusion

In summary, starting from a simple probabilistic generative model of neural oscillations, OCA provides a novel data-driven approach for analyzing multichannel synchronization within underlying oscillatory modes that are easier to interpret than conventional frequency-wise, cross-channel coherence. The overall approach adds significantly to existing methodologies for spatio-temporal decomposition by adding a formal representation of dynamics to the underlying generative model. The application of OCA on simulated and real M/EEG data demonstrates its capabilities as a principled dimensionality reduction tool that simultaneously provides a parametric description of the underlying oscillations and their activation pattern over the sensor array.

# Materials and methods

**Key resources table**

| Reagent type (species) or resource | Designation | Source or reference | Identifiers | Additional information |
|---|---|---|---|---|
| Software, algorithm | MNE-python 1.2 | https://mne.tools/stable/index.html; *Gramfort et al., 2014* | | |
| Software, algorithm | Eelbrain 0.37 | https://eelbrain.readthedocs.io/en/stable/; *Brodbeck et al., 2023* | | |
| Software, algorithm | purdonlabmeeg | This paper; *Das, 2024* | | Available at https://github.com/proloyd/purdonlabmeeg |

### M/EEG preprocessing

All prepossessing were performed using MNE-python 1.2 (*Gramfort et al., 2014*) and Eelbrain 0.37 (*Brodbeck et al., 2023*), with default setting of the respective functions.

#### EEG recording during propofol-induced unconsciousness

The EEG was bandpass filtered between 0.1 Hz to 40Hz, followed by downsampling to 100 Hz and average referencing prior to performing OCA. The 0.1 Hz highpass filtering was done to filter out slow drifts in the EEG recordings, which can adversely affect the OCA fitting procedure. All selected epochs had peak-to-peak signal amplitude less than 1000 mV. The prior on the noise-covariance was estimated from the same EEG recording, after high-pass filtering above 30 Hz.

#### Sleep EEG recording

The EEG data was first downsampled to 100 Hz after bandpass filtering within 1–40 Hz. The flat and noisy channels are first identified upon visual inspection. We computed neighborhood correlation for the rest of the channels and marked channels with median correlation, $\rho < 0.4$ as bad channels. These channels are dropped for the subsequent analysis.

#### MEG recording from the Human Connectome Project

We divided the entire resting-state MEG recording from subject 104 012_MEG, session 3 into 5 s epochs, and selected 34 epochs with peak-to-peak amplitude less than 4000 fT for OCA. Prior to fitting OCA, we used signal-space projection (*Uusitalo and Ilmoniemi, 1997*) to remove heartbeat and eye movement artifacts from MEG signals. The repaired MEG recordings are then downsampled to 100 Hz after applying appropriate anti-aliasing filter.

## Code availability

An implementation of the state inference and parameter learning algorithms described in this article is available as part of purdonlabmeeg Python library on GitHub at https://github.com/proloyd/purdon-labmeeg, copy archived at *Das, 2024* under MIT license. The simulation script to generate *Figure 2*, included in the repository, demonstrates how to run OCA on preprocessed EEG/MEEG data and visualize the results. Codes for analyzing experimental EEG and MEG recordings exactly follow the simulation script, i.e., is essentially a single function call with the experimental data (preprocessed according to previous section) as an argument and the results are to also be visualized as done in the simulation script. Thus they were not included as a separate repository, but can be made available to individuals upon request.

## Acknowledgements

This work was supported by the National Institutes of Health (grant no. R01AG054081-01A1) and Tiny Blue Dot Foundation. As requested by the Human Connectome Project, the following text is copied verbatim: MEG data in sSection 'Resting -state MEG recording' 'were provided by the Human Connectome Project, WU-Minn Consortium (Pprincipal Iinvestigators: David Van Essen and Kamil Ugurbil; 1U54MH091657) funded by the 16 NIH Institutes and Centers that support the NIH Blueprint for Neuroscience Research; and by the McDonnell Center for Systems Neuroscience at Washington University.

## Additional information

### Funding

| Funder | Grant reference number | Author |
|---|---|---|
| National Institutes of Health | R01AG054081-01A1 | Patrick L Purdon |
| Tiny Blue Dot Foundation | | Patrick L Purdon |

The funders had no role in study design, data collection and interpretation, or the decision to submit the work for publication.

### Author contributions

Proloy Das, Conceptualization, Resources, Data curation, Software, Formal analysis, Validation, Investigation, Visualization, Methodology, Writing - original draft, Writing – review and editing; Mingjian He, Resources, Data curation, Visualization; Patrick L Purdon, Conceptualization, Resources, Supervision, Funding acquisition, Investigation, Methodology, Project administration, Writing – review and editing

### Author ORCIDs

Proloy Das (iD) https://orcid.org/0000-0002-8807-042X
Mingjian He (iD) http://orcid.org/0000-0002-6688-8693
Patrick L Purdon (iD) https://orcid.org/0000-0003-0080-3340

Reviewer #1 (Public Review): https://doi.org/10.7554/eLife.97107.3.sa1
Author response https://doi.org/10.7554/eLife.97107.3.sa2

## Additional files

### Supplementary files
• MDAR checklist

## Data availability

The current manuscript is a computational study, so no data have been generated for this manuscript. An implementation of the state inference and parameter learning algorithms described in this article is available as part of purdonlabmeeg Python library on GitHub at https://github.com/proloyd/purdonlabmeeg (copy archived at *Das, 2024*) under MIT license. The simulation script to generate Figure 2, included in the repository, demonstrates how to run OCA on preprocessed EEG/MEEG data and visualize the results. Codes for analyzing experimental EEG and MEG recordings exactly follow the simulation script, i.e., is essentially a single function call with the experimental data (preprocessed according to previous section) as an argument and the results are to also be visualized as done in the simulation script. Thus they were not included as a separate repository, but can be made available to individuals upon request.

The following previously published dataset was used:

| Author(s) | Year | Dataset title | Dataset URL | Database and Identifier |
|---|---|---|---|---|
| Van Essen DC, Smith SM, Barch DM, Behrens TEJ, Yacoub E, Ugurbil K | 2013 | MEG data in Section Resting state MEG recording | https://db.humanconnectome.org | ConnectomeDB, 104012MEG |

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

# Appendix 1

## Mechanistic origin

Here, we detail how the multichannel multi-oscillator state-space model can be understood in terms of a biophysical model of brain wave generation.

## Charge-continuity and electromagnetic brain waves

Maxwell's equation provides the following charge-continuity equation that governs the spontaneous spatial-temporal evolution of the potential, $\Phi$, in brain tissues:

$$\frac{\partial}{\partial t}\nabla^2\Phi = -\nabla\cdot\Sigma\cdot\Phi. \tag{A1.1}$$

This differential equation is satisfied by traveling waves of the form, $\Phi \sim \exp\left[-j\left(\mathbf{k}\cdot\mathbf{r}+\Omega t\right)\right]$, where $\mathbf{r}$ and $t$ denote the spatial coordinate and time, respectively, with the wave number, $\mathbf{k}$, and the complex frequency, $\Omega$, satisfying following dispersion relation: $D(\Omega,\mathbf{k}) = j\Omega|k|^2 - \Sigma_{ij}k_ik_j - j\partial_i\Sigma_{ij}k_j$, in tensor notation. Separating the real and imaginary parts of the complex frequency, $\Omega = \omega + j\gamma$, we arrive at the following expressions for decaying and oscillatory frequencies:

$$\gamma = -\frac{\Sigma_{ij}k_ik_j}{|\mathbf{k}|^2}, \qquad \omega = \frac{\partial_i\Sigma_{ij}k_j}{|\mathbf{k}|^2}. \tag{A1.2}$$

Clearly, wave solutions $\Phi$ satisfying $|\gamma/\omega| \ll 1$, that is, the oscillatory frequency is much larger than the decaying frequency, can give rise to electromagnetic oscillations in brain parenchyma, governed by the brain composition and architecture. Detailed treatment of such a model, considering the anisotropy and inhomogeneity of the brain tissue, can be found in *Galinsky and Frank, 2020*.

## Traveling wave solution to the state-space model

Here, we consider one such oscillatory electric potential being recorded at $L$ sensors placed at coordinates $\mathbf{r}_1, \mathbf{r}_2, \cdots, \mathbf{r}_L$ (w.r.t. a reference electrode placed at infinity). The potentials sampled (sampling frequency, $f_s = 1/\Delta t$) at those electrodes at time index $n$ are given as

$$\Phi(l,n) = \Re\left\{A_l\exp\left[-j\left(\mathbf{k}\cdot\mathbf{r}_l + \Omega\, n\Delta t\right)\right]\right\} \tag{A1.3}$$

with $A_l$ representing the amplitude observed at $l$th electrode. The separability of the spatial part $\phi^{spatial}(l) = A_l\exp\left[-j\left(\mathbf{k}\cdot\mathbf{r}_l\right)\right]$ and the temporal part $\Phi^{temporal}(n) = \exp\left[-j\left(\Omega\, n\Delta t\right)\right]$ of the general solution is crucial for the following development since only the temporal part, $\Phi^{temporal}(n+1)$, evolves in time,

$$\Phi^{temporal}(n+1) = \exp\left[j\Omega\Delta t\right]\Phi^{temporal}(n), \tag{A1.4}$$

while the spatial part $\phi^{spatial}(l)$ remains constant. Adapting the following real-valued matrix-vector notation that consists of the real and imaginary parts of the temporal and spatial components, $\Phi^{temporal}(n) = x_{n,1} + jx_{n,2}$ and $\Phi^{spatial}(l) = m_{l,1} + jm_{l,2}$, we arrive at:

$$\begin{bmatrix} x_{n+1,1} \\ x_{n+1,2} \end{bmatrix} = \exp\left[-\gamma\Delta t\right]\begin{bmatrix} \cos(\omega\Delta t) & -\sin(\omega\Delta t) \\ \sin(\omega\Delta t) & \cos(\omega\Delta t) \end{bmatrix}\begin{bmatrix} x_{n,1} \\ x_{n,2} \end{bmatrix}, \tag{A1.5}$$

$$y_{n+1}^{(l)} = \begin{bmatrix} A_lm_{l,1} & A_lm_{l,2} \end{bmatrix}\begin{bmatrix} x_{n+1,1} \\ x_{n+1,2} \end{bmatrix}, \tag{A1.6}$$

where $y_{n+1}^{(l)}$ is the recorded potential at electrode $l$ at time index $(n+1)$. Essentially, the complex-valued oscillatory function, $\Phi^{temporal}(n)$, is an analytic signal whose real and imaginary parts constitute the ordered pair, $\mathbf{x}_n = [x_{n,1}, x_{n,2}]$, which we call the *oscillation state*. *Equation A1.5* describes a circular motion of the oscillation state $\mathbf{x}_n$ in a 2D state space with frequency $\omega$ at every time step through the damped rotation matrix $\exp\left[-\gamma\Delta t\right]\mathcal{R}(\omega\Delta t)$, while a projection, dictated by the electrode location (*Equation A1.6*), is recorded at the EEG electrodes as an oscillation (see *Figure 1a*). This deterministic

model forms the basis for the probabilistic linear Gaussian state-space model in *Equation 1* which introduces process and observation noise, $\boldsymbol{v}_n$ and $\epsilon_n$, respectively, with redefined spatial components as $c_{l,*} = A_l m_{l,*}$, and the decay term $\exp\left[-\gamma\Delta t\right]$ as a damping parameter, $a$. We also represented the frequency of oscillation, $f$, as $f = \omega/2\pi$ by replacing $\omega\Delta t = 2\pi f/f_s$.

## Appendix 2

## Oscillation states and analytic signals

Here, we consider real-valued observations, $y_n \in \mathbb{R}$, admitting a state-space representation in complex plane, that is, that is generated from complex-valued states, $z_n \in \mathbb{C}$:

$$z_n = \rho x_{n-1} + q_n$$
$$y_n = \frac{1}{2}(z_n + \bar{z}_n) + r_n \tag{A2.1}$$

where $\bar{z}_n$ denotes complex conjugate of $\bar{z}_n$, $q_n$ and $r_n$ are complex-valued and real-valued random variables. In steady state, that is, $t \to \infty$, the states, $z_n$, admit Fourier representation given by

$$Z(e^{-j\omega}) = \frac{Q(e^{-j\omega})}{1 - \rho e^{-j\omega}} \tag{A2.2}$$

It is clear that if $q_n$ is an analytic signal, that is, $Q(e^{-j\omega})$ has no negative frequency component, the same will be true for $Z(e^{-j\omega})$.

Further, since $q_n$ is an analytic signal, there exists a real-valued signal $s_n$, such that $q_n = s_n + j(h * s)_n$, where discrete time Fourier transform of $h_n$ is given by

$$H(e^{-j\omega}) = \begin{cases} -j & \text{for } 0 < \omega \leq \pi \\ 0 & \text{for } \omega = 0 \\ j & \text{for } -\pi < \omega < 0 \end{cases} . \tag{A2.3}$$

Considering $s_n$ to be a white noise with variance $\sigma^2$, it is easy to verify that $h_n * s_n$ is also white noise with variance $\sigma^2$ and uncorrelated to $s_n$. This makes $q_n$ to be a 'analytic' white noise, that is power spectrum density of $q_n$ is flat for $0 > \omega > \pi$, and zero in $-\pi > \omega > 0$. We can then derive the following state-space representation, with real valued but 2D states as in **Equation 1**, but with additional constraint that $v_{n,1}$ and $v_{n,2}$ are related by $v_{n,2} = (h * v_1)_n$. But since such constraint is not easy to enforce during the inference or model learning from experimental data, so we relaxed the constraint on the state-noise covariance to be the one stated in **Equation 1**.

## Extraction of instantaneous amplitude and phase

However, we can still compute the instantaneous amplitude and phase of the signal from this state-space representation from the following observation. The oscillation states assume their successive values in a way that traces limit cycle-like maps around origin (see **Figure 1a**). This allows us to define the instantaneous phase as the angle the oscillation state makes from a fixed reference direction at any given timepoint (**Rosenblum et al., 1997**, section A.1 and 2). For simplicity, we consider the positive direction on the real line as the reference. The amplitude is simply given by the distance from the origin.

$$A_t^{(*)} = \sqrt{x_{t,1}^{(*)^2} + x_{t,2}^{(*)^2}} \qquad P_t^{(*)} = \arctan\left(\frac{x_{2,t}^{(*)}}{x_{1,t}^{(*)}}\right) \tag{A2.4}$$

In this sense, the oscillator state-space representation provides a generalization of phasor concept, similar to analytic signals (i.e., by allowing for time-varying amplitude, phase, and frequency, in contrast to invariant amplitude, phase, and frequency of phasor) but with relatively relaxed constraint on the relation between real and imaginary parts.

## Appendix 3

## Model parameter estimation and model selection

### Negative variational free energy

For an $L$-channel M/EEG recording $\mathbf{y}_t$, $t = 1, 2, \cdots, T$, $M$-oscillator probabilistic state-space oscillator model admits to the following distribution:

$$p(\{\mathbf{x}_t, \mathbf{y}_t\} \mid \mathbf{C}, \mathbf{R}, \mathbf{F}, \mathbf{Q}, M) = \prod_{t=1}^{T} \frac{1}{\sqrt{|(2\pi)\mathbf{R}|}} \exp{-\frac{1}{2} \|\mathbf{y}_t - \mathbf{C}\mathbf{x}_t\|_{\mathbf{R}^{-1}}^2}$$
$$\frac{1}{\sqrt{|(2\pi)\mathbf{Q}|}} \exp{-\frac{1}{2} \|\mathbf{x}_t - \mathbf{F}\mathbf{x}_{t-1}\|_{\mathbf{Q}^{-1}}^2} \tag{A3.1}$$

Since $\mathbf{F}$ and $\mathbf{Q}$ are parameterized using hyperparameter $\boldsymbol{\theta}$, we will re-parameterize left-hand side of *Equation A3.1* as $p\left(\{\mathbf{x}_t, \mathbf{y}_t\} \mid \mathbf{C}, \mathbf{R}, \boldsymbol{\theta}, M\right)$. To simplify our exposition, we define $\mathbf{X} = [\mathbf{x}_1, \mathbf{x}_2, \cdots, \mathbf{x}_N]$ and use $\mathbf{X}$ and $\{\mathbf{x}_t\}$ interchangeably (similarly for $\mathbf{Y}$). We also extensively use $vec(\circ)$ notation and associated terminology introduced in *Muirhead, 1982*.

We note that computation of ensemble likelihood of the presented model requires marginalizing over $\mathbf{X}, \mathbf{C}, \mathbf{R}$, and $M$ given the priors in *Equations 4* and *6*:

$$
\begin{aligned}
p(\{\mathbf{y}_t\}) \quad &= \sum_{M=1}^{M_{max}} \int p(\mathbf{X}, \mathbf{Y}, \mathbf{C}, \mathbf{R} \mid \alpha, \boldsymbol{\theta}, M)\, p(M)\, d\{\mathbf{X}, \mathbf{C}, \mathbf{R}\} \\
&= \sum_{M=1}^{M_{max}} \int p(\mathbf{X}, \mathbf{Y} \mid \mathbf{C}, \mathbf{R}, \alpha, \boldsymbol{\theta}, M)\, p(\mathbf{C} \mid \alpha, M)\, p(\mathbf{R})\, p(M)\, d\{\mathbf{X}, \mathbf{C}, \mathbf{R}\}
\end{aligned}
$$

However, this involves an intractable integration that cannot be performed analytically. We avoid this intractability by using a lower bound to the ensemble likelihood as a surrogate for the same. Specifically, we invoke Neal–Hinton representation theorem (*Neal and Hinton, 1998*) to obtain a lower bound on the ensemble log-likelihood as

$$
\begin{aligned}
\mathcal{L} \quad &= \log p\left(\{\mathbf{y}_t\}\right) \\
&= \log \sum_{M=1}^{M_{max}} \int p\left(\mathbf{X}, \mathbf{Y}, \mathbf{C}, \mathbf{R} \mid \alpha, \boldsymbol{\theta}, M\right) p(M) d\{\mathbf{X}, \mathbf{C}, \mathbf{R}\} \\
&= \log \sum_{M=1}^{M_{max}} \int \frac{p\left(\mathbf{X}, \mathbf{Y}, \mathbf{C}, \mathbf{R} \mid \alpha, \boldsymbol{\theta}, M\right) p(M)}{q\left(\mathbf{X}, \mathbf{C}, \mathbf{R} \mid M\right) q(M)} q\left(\mathbf{X}, \mathbf{C}, \mathbf{R} \mid M\right) q(M) d\{\mathbf{X}, \mathbf{C}, \mathbf{R}\} \\
&\geq \sum_{M=1}^{M_{max}} q(M) \int \log \frac{p\left(\mathbf{X}, \mathbf{Y}, \mathbf{C}, \mathbf{R} \mid \alpha, \boldsymbol{\theta}, M\right) p(M)}{q\left(\mathbf{X}, \mathbf{C}, \mathbf{R}, \mid M\right) q(M)} q\left(\mathbf{X}, \mathbf{C}, \mathbf{R} \mid M\right) d\{\mathbf{X}, \mathbf{C}, \mathbf{R}\} \\
&= \sum_{M=1}^{M_{max}} q(M) \left[\log \frac{p(M)}{q(M)} + \int \log \frac{p\left(\mathbf{X}, \mathbf{Y}, \mathbf{C}, \mathbf{R}, \alpha, \boldsymbol{\theta} \mid M\right)}{q\left(\mathbf{X}, \mathbf{C}, \mathbf{R} \mid M\right)} q\left(\mathbf{X}, \mathbf{C}, \mathbf{R} \mid M\right) d\{\mathbf{X}, \mathbf{C}, \mathbf{R}\}\right] \\
&= \left\langle \log \frac{p(M)}{q(M)} + \left\langle \log \frac{p\left(\mathbf{X}, \mathbf{Y}, \mathbf{C}, \mathbf{R}, \alpha, \boldsymbol{\theta} \mid M\right)}{q\left(\mathbf{X}, \mathbf{C}, \mathbf{R} \mid M\right)} \right\rangle_{q(\mathbf{X}, \mathbf{C}, \mathbf{R} \mid M)} \right\rangle_{q(M)} := \mathcal{F},
\end{aligned}
\tag{A3.2}
$$

which holds for any arbitrary conditional distribution $q$ and can be computed analytically if $q$ is chosen carefully. This lower bound is known as *negative variational free energy* (*Quinn and Šmídl, 2006*) and attains the actual ensemble log-likelihood only when $q$ is the exact Bayes posterior distribution (*Attias, 1999*). In the last line, we use $\langle \circ \rangle_{q(\mathbf{X}, \mathbf{C}, \mathbf{R} \mid M)}$ to denote the average of the expression within $\langle \rangle$ with respect to the model posterior $q(\mathbf{X}, \mathbf{C}, \mathbf{R} \mid M)$.

Since negative variational free energy as a lower bound on the ensemble log-likelihood, maximization of negative variational free energy will tend to maximize of the ensemble log-likelihood. Even though global maximization of the ensemble log-likelihood is not guaranteed, the ensemble log-likelihood at the point of maximum negative variational free energy is guaranteed to at least greater than the negative variational free energy. On the other hand, the negative variational free energy can be shown to be proportional to the negative Kullback–Leibler divergence between the exact posterior and approximate posterior (*Quinn and Šmídl, 2006*). So, the maximization shall result in closer approximation exact posterior. In the next section, we work with the intuition that maximization of negative variational free energy leads to better approximation to the exact posterior and increment of ensemble log-likelihood.

## Variational Bayes inference

However, the fact that the ensemble log-likelihood cannot be found analytically makes computation of the exact Bayes posterior distribution intractable. We thus employ an efficient VB inference procedure that approximates the exact Bayes posterior with a distribution of form:

$$q\left(\{\mathbf{x}_t\}, \mathbf{C}, \mathbf{R} \mid M\right) = q\left(\{\mathbf{x}_t\} \mid M\right) q\left(\mathbf{C} \mid M\right) q\left(\mathbf{R} \mid M\right),$$

(A3.3)

that is, that factorizes over $\{\mathbf{x}_t\}, \mathbf{C}, \mathbf{R}$, conditional on the model structure, $M$. This particular choice of functional constraint leads to tractable conditional distributions when the trailing term of *Equation A3.2* is subjected to maximization w.r.t. functional form:

$$q\left(\mathbf{X} \mid M\right) = \frac{1}{\sqrt{|(2\pi)\,\Sigma_x|}} \exp -\frac{1}{2}\left\|vec(\mathbf{X}) - \mu_x\right\|_{\Sigma_x^{-1}}^2$$

(A3.4)

$$q\left(\mathbf{C} \mid M\right) = \frac{1}{\sqrt{|(2\pi)\,\Sigma_c|}} \exp -\frac{1}{2}\left\|vec(\mathbf{C}) - \mu_c\right\|_{\Sigma_c^{-1}}^2$$

(A3.5)

$$q\left(\mathbf{R} \mid M\right) = \gamma_{\rho,L} |\Omega|^{\rho/2} |\mathbf{R}|^{-(\rho+L+1)/2} \exp -\frac{1}{2}\mathrm{Tr}\left\{\Omega\mathbf{R}^{-1}\right\}$$

(A3.6)

Here, we omit the trivial derivation of the functional form for brevity of the presentation; curious readers are encouraged to expand the trailing term of *Equation A3.2* and follow *Attias, 1999*, section 2.4 as an exercise to obtain the optimal functional forms.

Once the abovementioned optimal functional forms of the approximate posterior distribution have been identified, the next task is to find the optimal quantities that parameterize those functional forms. We show that the parameters of any of these distribution can be expressed in terms of the M/EEG data and moments of other two distributions. Generally speaking, the expressions can be found by collecting relevant terms in the expansion of trailing term of *Equation A3.2*, followed by simple algebraic manipulation. For example, the parameters of $q(\mathbf{R} \mid M)$ are given by

$$\Omega = \Psi + \sum_{t=1}^{T}\langle \mathbf{y}_t - \mathbf{C}\mathbf{x}_t\rangle_{q(\mathbf{X},\mathbf{C}|M)} \text{ and } \rho = \nu + L.$$

(A3.7)

Expression for the parameters of the Gaussian distribution is a bit involved if one attempts collecting terms and try completing squares. Instead, we use the following facts about Gaussian log-posteriors: (1) since the mode and mean of a multivariate Gaussian distribution coincide, the mean of Gaussian distributions can be found by maximizing the approximate log-posterior, and (2) the negative Hessian of the log-posterior is the inverse of covariance matrix (*Fahrmeir, 1992*). It is worth mentioning here that the log-posterior maximization can be carried out in closed form by solving a high-dimensional system of linear equations (first-order optimality criterion, *Boyd and Vandenberghe, 2004*), which involves Hessian matrix inversion.

In case of the latent oscillator states, $q(\mathbf{X} \mid M)$, the expressions of inverse covariance matrix and the mean vector is as follows:

$$\Sigma_x^{-1} = \begin{bmatrix} \mathbf{D} & \mathbf{S} & \mathbf{0} & \mathbf{0} & \cdots & \mathbf{0} & \mathbf{0} \\ \mathbf{S}^\top & \mathbf{D} & \mathbf{S} & \mathbf{0} & \cdots & \mathbf{0} & \mathbf{0} \\ \mathbf{0} & \mathbf{S}^\top & \mathbf{D} & \mathbf{S} & \cdots & \mathbf{0} & \mathbf{0} \\ \vdots & \vdots & \vdots & \vdots & \vdots & \vdots & \vdots \\ \mathbf{0} & \mathbf{0} & \mathbf{0} & \mathbf{0} & \cdots & \mathbf{S}^\top & \mathbf{D} \end{bmatrix}$$

$$\mu_x = \Sigma_x \times vec\left(\mathbf{B}\mathbf{y}_t\right)$$

where

$$\mathbf{D} = \left\langle \mathbf{C}^\top\mathbf{R}^{-1}\mathbf{C}\right\rangle_{q(\mathbf{C},\mathbf{R}|M)}$$
$$+\widehat{\mathbf{F}}^\top\widehat{\mathbf{Q}}^{-1}\widehat{\mathbf{F}} + \widehat{\mathbf{Q}}^{-1}$$
$$\mathbf{S} = -\widehat{\mathbf{F}}^\top\widehat{\mathbf{Q}}^{-1}$$

$$\mathbf{B} = \left\langle \mathbf{C}^\top\mathbf{R}^{-1}\right\rangle_{q(\mathbf{C},\mathbf{R}|M)}$$

(A3.8)

We exploit the block tridiagonal structure of the hessian matrix to realize a fast and stable inverse (**Asif and Moura, 2000**; **Jain et al., 2007**). This facilitates fast computation of the mean and covariance matrix of the approximate log-posterior.

Before tackling the case of the mixing matrix, $\mathbf{C}$, we note that each latent oscillator state is a pair of two coordinates and these two oscillator coordinates correspond to two consecutive columns of the mixing matrix, that is, contribution of any given latent oscillator state on the observation can be viewed as the inner product, that is, $\mathbf{y}_t^{(l)} = \mathbf{c}_{l,m}\mathbf{x}_t^{(m)}$, so that the column pairs $\left(\mathbf{C}_{:,2l-1:2l}\right)$ and latent oscillator states $\left(\mathbf{X}_{2l-1:2l,:}\right)$ are only unique up to a scaling and a rotation. We resolve this ambiguity by simply fixing the first row of the mixing matrix column pair to $\begin{bmatrix} 1 & 0 \end{bmatrix}$ for all the oscillators. With this modification, the parameters of $q(\mathbf{C}\mid M)$ are given as

$$
\begin{aligned}
\boldsymbol{\Sigma}_c^{-1} &= \boldsymbol{\Lambda}_{2:L,2:L} \otimes T\mathbf{P}_{xx} + \alpha\mathbf{I}_{2(L-1)M\times 2(L-1)M} \\
\boldsymbol{\mu}_c &= \boldsymbol{\Sigma}_c\left(T\mathrm{vec}\left(\left(\boldsymbol{\Lambda}_{2:N}\mathbf{P}_{yx}\right)^\top\right) - \boldsymbol{\lambda}_1 \otimes \left(\mathbf{P}_{xx}\mathbf{m}_1^\top\right)\right)
\end{aligned}
\tag{A3.9}
$$

where

$$
\Lambda = \left\langle \mathbf{R}^{-1} \right\rangle_{q(\mathbf{R}|M)}, \quad \mathbf{P}_{xx} = \frac{1}{T}\sum_{t=1}^{T} \left\langle \mathbf{x}_t\mathbf{x}_t^\top \right\rangle_{q(\mathbf{X}|M)}, \quad \mathbf{P}_{yx} = \frac{1}{T}\sum_{t=1}^{T} \left\langle \mathbf{y}_t\mathbf{x}_t^\top \right\rangle_{q(\mathbf{X}|M)}.
$$

Using the properties of Kronecker products (**Muirhead, 1982**), the inversion of the negative Hessian matrix and solution of the system of the linear equations can be carried out in a fast and efficient manner.

Evidently, this interconnectedness of the distributional parameters of the marginals of $\{\mathbf{x}_t\}, \mathbf{C}, \mathbf{R}$, given model structure $M$, is a testament to the inter-dependence of these three variables. Since joint posterior distribution of is assumed to factorize over the marginals, that is, $\{\mathbf{x}_t\}, \mathbf{C}, \mathbf{R}$, given model structure $M$ and observations $\mathbf{Y}$ are assumed to be independent, inter-dependence has been captured by the interrelated distributional parameters. In fact, the interrelated parameters pass information among them using the sufficient statistic (SS) of these distributions that are required to compute the averages in the update equations. The VB approach for posterior inference therefore leads to an iterative scheme for each model structure, $M$, when the hyperparameters $(\boldsymbol{\theta}, \alpha)$ are specified: we cyclically update the marginal posterior parameters of $\{\mathbf{x}_t\}, \mathbf{C}, \mathbf{R}$ according to **Equations A3.8, A3.9, and A3.7**, , respectively. Each time a distribution (i.e., its parameters) is updated, the SS associated with the distribution is also recomputed to pass to the next update. The iterations are repeated until a stopping criteria is met: stopping criteria could be a fixed number of iterations or when the objective, that is, variational free energy stabilizes. The marginal posterior parameters obtained at the last iteration are the optimal distributional parameters providing the VB inference.

## Generalized EM algorithm

The VB inference of form presented in **Equation A3.3** still requires the hyperparameters $\boldsymbol{\theta}$ and $\alpha$ to be specified. The hyperparameters that maximize the negative variational free energy are the second best choice for the hyperparameters, right after the ones that maximize the ensemble likelihood. Since the direct maximization w.r.t. the hyperparameters is impractical, we use an generalized version of expectation maximization (EM) algorithm that uses the variational posterior inference instead of the exact posterior to compute the expectation of the complete log-likelihood expression (**Dempster et al., 1977**; **Fahrmeir, 1992**; **Shumway and Stoffer, 1982**) to learn these hyperparameters from the M/EEG recording. One requires the following averages computed w.r.t. the approximate posterior $q(\mathbf{X}\mid M)$,

$$
\begin{aligned}
\mathbf{U}^{(m)} &= \sum_{t=1}^{T} \left\langle \mathbf{x}_{t-1}^{(m)}\mathbf{x}_{t-1}^{(m)^\top} \right\rangle_{q(\mathbf{X}|M)}, \\
\mathbf{V}^{(m)} &= \sum_{t=1}^{T} \left\langle \mathbf{x}_t^{(m)}\mathbf{x}_{t-1}^{(m)^\top} \right\rangle_{q(\mathbf{X}|M)}, \\
\mathbf{W}^{(m)} &= \sum_{t=1}^{T} \left\langle \mathbf{x}_t^{(m)}\mathbf{x}_t^{(m)^\top} \right\rangle_{q(\mathbf{X}|M)},
\end{aligned}
$$

to update the hyperparameters $\boldsymbol{\theta}^{(m)} = \left(f^{(m)}, a^{(m)}, (\sigma^2)^{(m)}\right)$ as follows:

$$f^{(m)} = \frac{1}{2\pi} \arctan \left\{ \frac{\mathrm{rt}\left\{\boldsymbol{V}^{(m)}\right\}}{\mathrm{tr}\left\{\boldsymbol{V}^{(m)}\right\}} \right\}, \tag{A3.10}$$

$$a^{(m)} = \frac{\sqrt{\mathrm{tr}\left\{\mathbf{V}^{(m)}\right\}^2 + \mathrm{rt}\left\{\mathbf{V}^{(m)}\right\}^2}}{\mathrm{tr}\left\{\mathbf{U}^{(m)}\right\}} \tag{A3.11}$$

$$\sigma^{2(m)} = \frac{1}{2\pi}\left(\mathrm{tr}\left\{\mathbf{W}^{(m)}\right\} - a^{(m)2}\mathrm{tr}\left\{\mathbf{U}^{(m)}\right\}\right) \tag{A3.12}$$

On the other hand, α update takes a relatively simple form:

$$\alpha = \frac{2(L-1)M}{\left\langle \|\mathbf{C}_{2:L,:}\|^2 \right\rangle_{q(\mathbf{C}|M)}}. \tag{A3.13}$$

In brief, we start with an initial set of the hyperparameters $\theta$ and $\alpha$, and keep alternating between variational posterior inference (E-step) and hyperparameter update (M-step) until the hyperparameters or the variational free energy stabilize. Once the suitable hyperparameters are obtained, the variational inference is run for the last time, providing us with the *final* oscillation components.

## Evaluation of variational free energy

Considering update rules of *Equations A3.7, A3.13*, , and *Equation A3.10* we can simplify the second term in *Equation A3.2* as

$$\left\langle \log \frac{p\left(\mathbf{X}, \mathbf{Y}, \mathbf{C}, \mathbf{R}, \alpha, \boldsymbol{\theta} \mid M\right)}{q\left(\mathbf{X}, \mathbf{C}, \mathbf{R} \mid M\right)} \right\rangle_{q(\mathbf{X}, \mathbf{C}, \mathbf{R}|M)} = -\frac{(L+2M)T}{2}\log 2\pi$$
$$-\frac{T}{2}\log|\mathbf{Q}| - \frac{2M(L-1)}{2} - \left\langle \log q\left(\mathbf{X} \mid M\right)_{q(\mathbf{X}|M)} \right\rangle$$
$$+\frac{2M(L-1)}{2}\log\frac{\alpha}{2\pi} - \frac{2MT}{2} - \left\langle \log q(\mathbf{M} \mid m) \right\rangle$$
$$+\frac{\nu}{2}\log|\boldsymbol{\Psi}| + \gamma_{\nu,n} - \left(\frac{\nu+T}{2}\log|\boldsymbol{\Omega}| + \gamma_{\nu+T,n}\right). \tag{A3.14}$$

After noting the following property of Gaussian random variable, $\mathbf{Z} \sim \mathcal{N}_p(\boldsymbol{\mu}_z, \boldsymbol{\Sigma}_z)$:

$$\langle \log p(\mathbf{Z})\rangle_{p(\mathbf{Z})} = -\frac{1}{2}\log\left|(2\pi)\boldsymbol{\Sigma}_z\right| - \frac{p}{2}$$

*Equation A3.14* can be further simplified as:

$$\left\langle \log \frac{p\left(\mathbf{X}, \mathbf{Y}, \mathbf{C}, \mathbf{R}, \alpha, \boldsymbol{\theta} \mid M\right)}{q\left(\mathbf{X}, \mathbf{C}, \mathbf{R} \mid M\right)} \right\rangle_{q(\mathbf{X}, \mathbf{C}, \mathbf{R}|M)} = -\frac{LT}{2}\log 2\pi - \frac{T}{2}\log|\mathbf{Q}| + \frac{1}{2}\log|\boldsymbol{\Sigma}_x|$$
$$+\frac{1}{2}\log|\alpha\boldsymbol{\Sigma}_c| + \frac{\nu}{2}\log|\boldsymbol{\Psi}| \quad +\gamma_{\nu,n} - \left(\frac{\nu+T}{2}\log|\boldsymbol{\Omega}| + \gamma_{\nu+T,n}\right) \tag{A3.15}$$

We note here that the log-determinant of $\boldsymbol{\Sigma}_x$ could be computed while solving *Equation A3.8* exploiting the tridiagonal structure of $\Sigma_x{}^{-1}$. Similarly, the log-determinant of $\boldsymbol{\Sigma}_c$ could be computed while solving *Equation A3.9* using the properties of Kronecker products. Matrices $\boldsymbol{\Phi}$ and $\boldsymbol{\Omega}$ are in the sensor space dimension, thus log-determinant computation is relatively cheaper. The remaining matrix $\mathbf{Q}$ is a diagonal matrix, thus its log-determinant can be computed easily. As a whole, the variational free energy for model structure $M$ can be computed as a by-product of the iterative updates.

## Initialization of parameters and hyperparameters

Because the ensemble log-likelihood (hence the negative variational free energy) is not concave in general, initial values for the parameters and hyperparameters must be carefully chosen. We use the interpretation of the MVAR processes as a superimposition of multiple oscillatory components (*Matsuda and Komaki, 2017b*; *Quinn et al., 2021*; *Neumaier and Schneider, 2001*).

First of all, we estimate $\boldsymbol{\Psi}$ using the fact that for single-channel noisy recording, $y^{(l)}$ where $S_{y^{(l)}y^{(l)}}(f)$ denotes the PSD of the signal:

$$10 \log_{10} \frac{R}{f_s} = \lim_{f \to \infty} S_{y^{(l)}y^{(l)}}(f). \tag{A3.16}$$

We generalize *Equation A3.16* to multichannel recording by high-pass filtering the M/EEG recording at $f_s - \Delta f$, $(\Delta f > 0)$, and estimating inverse-Wishart parameters from the sample covariance of the high-pass filtered data, $\widetilde{\mathbf{y}}_t$. We set $\nu = T$, and $\boldsymbol{\Psi} = \sum_{t=1}^{T} \mathbf{x}_t \mathbf{x}_t^\top$. We also initialize the noise covariance matrix as $\mathbf{R} = \sum_{t=1}^{T} \mathbf{x}_t \mathbf{x}_t^\top / T$.

We then fit an MVAR model of order $p$ on the multichannel data. Order $p$ is chosen according to Akaike information criteria (*Cavanaugh and Neath, 2019*). We perform eigendecomposition of the companion form of the MVAR parameter matrix (*Neumaier and Schneider, 2001*) to yield $\boldsymbol{\lambda}$, $\mathbf{V}$, and $\mathbf{W}$ as eigenvalues, right and left eigenvectors. We choose the eigenvalues with $\Im\{\lambda^{(m)}\} > 0$ and corresponding right eigenvectors $\mathbf{w}^{(m)}$ and collect them as $\widehat{\boldsymbol{\lambda}}$ and $\widehat{\mathbf{W}}$ (assume there are $M$ such eigenvalues). The frequency and damping parameters are initialized as

$$f^{(m)} = \arg\left(\lambda^{(m)}\right) \frac{f_s}{2\pi} \qquad a^{(m)} = \left|\lambda^{(m)}\right|.$$

For $\sigma^{(1)2}, \sigma^{(m)2}, \cdots, \sigma^{(M)2}$, we use the linear combination of the theoretical PSD of the oscillatory components that approximate the multitaper PSD (*Babadi and Brown, 2014*) of the leading time series the best. In short, we compute the multitaper PSD of the leading time series and the theoretical PSDs of the oscillatory components (*Soulat et al., 2022*, Supplementary Information) on the same frequency grid of spacing $f_s/(2K)$. The multitaper PSD, $\boldsymbol{\rho}$, is a $1 \times K$ vector, whereas the theoretical PSDs, $\phi^{(m)}$, form a $M \times K$ matrix $\boldsymbol{\Phi}$. We initialize the $\sigma^{(1)2}, \sigma^{(m)2}, \cdots, \sigma^{(M)2}$ as the solution of the following linear equation:

$$\begin{bmatrix} \sigma^{(1)2} & \sigma^{(m)2} & \cdots & \sigma^{(M)2} \end{bmatrix} \boldsymbol{\Phi} = \boldsymbol{\rho}. \tag{A3.17}$$

In order to select a subset of the discovered $M$ oscillatory components, we sort the oscillatory components (and columns of the $\widehat{\mathbf{W}}$) according to their theoretical contribution (most to least):

$$\sigma^{2(m)} \|\widehat{\mathbf{w}}^{(m)} / \widehat{w}_1^{(m)}\|_2 \sum_f \phi^{(m)}(f), \tag{A3.18}$$

and select the leading oscillatory parameters (and columns of the $\widehat{\mathbf{W}}$).

With *Equation A3.17*, the spatial mixing components for the leading time series become $\mathbf{c}_{l,m} = \begin{bmatrix} 1 & 0 \end{bmatrix}$. The rest of the elements of spatial mixing matrix, $\mathbf{C}$, are initialized from $\widehat{\mathbf{W}}$ as

$$\mathbf{c}_{l,m} = \begin{bmatrix} \Re\left(\dfrac{\widehat{\mathbf{w}}_l^{(m)}}{\widehat{w}_1^{(m)}}\right) & -\Im\left(\dfrac{\widehat{\mathbf{w}}_l^{(m)}}{\widehat{w}_1^{(m)}}\right) \end{bmatrix}$$

Lastly, $\alpha$ is initialized by $2(L-1)M / \|\mathbf{C}_{2:L}\|_{Fro}$ with the abovementioned $\mathbf{C}$.

## Model structure posterior

Finally, it is easy to verify that the following discrete distribution maximizes *Equation A3.2*:

$$q(M) = \frac{p(M)\exp - \left\langle \log \frac{p\left(\mathbf{X}, \mathbf{Y}, \mathbf{C}, \mathbf{R}, \alpha, \boldsymbol{\theta} \mid M\right)}{q\left(\mathbf{X}, \mathbf{C}, \mathbf{R} \mid M\right)} \right\rangle_{q(\mathbf{X},\mathbf{C},\mathbf{R}|M)}}{\sum_{M'=1}^{M_{max}} p(M')\exp - \left\langle \log \frac{p\left(\mathbf{X}, \mathbf{Y}, \mathbf{C}, \mathbf{R}, \alpha, \boldsymbol{\theta} \mid M'\right)}{q\left(\mathbf{X}, \mathbf{C}, \mathbf{R} \mid M'\right)} \right\rangle_{q(\mathbf{X},\mathbf{C},\mathbf{R}|M')}}, \tag{A3.19}$$

which can be computed readily once the final variational inferences are made for $M = 1, \cdots, M_{\max}$. The negative variational free energy of the modeling framework is given by

$$\mathcal{F} = \sum_{M'=1}^{M_{max}} p(M')\exp - \left\langle \log \frac{p\left(\mathbf{X}, \mathbf{Y}, \mathbf{C}, \mathbf{R}, \alpha, \boldsymbol{\theta} \mid M'\right)}{q\left(\mathbf{X}, \mathbf{C}, \mathbf{R} \mid M'\right)} \right\rangle_{q(\mathbf{X},\mathbf{C},\mathbf{R}|M')}. \tag{A3.20}$$

## Empirical Bayes inference and model selection

Since the model order is not known, we estimate the model order, $M$, by maximizing the approximate posterior distribution of model order, $q(M)$, and treat $M$ as if it were known to be equal to this estimate. The same procedure is followed for the hyperparameters, for which we set them to the values that maximize the negative variational free energy. This way of determining model order or estimating unknown parameters comes under the umbrella of empirical Bayes framework (*Robbins, 1964*; *Efron and Morris, 1973*; *Efron and Morris, 1975*; *Morris, 1983*). Briefly, it provides a first-order approximation to the posterior mean of these quantites and neglects the uncertainty (i.e., second-order statistics) of the estimated quantities.

# Appendix 4

## Comparison of OCA to traditional approaches in experimental EEG data

We compare OCA with an oscillation finding approach based on channel-wise PSD visualization and ICA in a subset of real human EEG recording, taken from *Figure 4* to provide empirical justification for OCA. For *Appendix 4—figure 1*, we used a 3.5-s-long EEG segment during the maintenance of 2 µg effect-site concentration of propofol. We expect strong frontal alpha activity and overall increase in slow activity as per existing literature (*Purdon et al., 2013*; *Cimenser et al., 2011*).

The frequency-domain approach examined here is based on the popular multitaper method (*de Cheveigné and Parra, 2014*): given a bandwidth parameter, power spectrum density is computed for individual channels and a butterfly plot is made (see *Appendix 4—figure 1A*, left panels). The oscillatory alpha activities are then identified visually (red overlay) as the peaks in these power spectrum plots, and their scalp distribution are obtained by averaging channel-wise power within given frequency bands around the peaks as shown in *Appendix 4—figure 1A*, right panels. We chose three different bandwidth, 1 Hz, 2 Hz, and 5 Hz, to demonstrate the inconsistency of frequency domain methods in 'producing' peaks in the line plots of the spectrum. Spectrum with 1 Hz bandwidth (top row) exhibits a lot of spurious peaks, whereas spectrum with 5 Hz bandwidth (bottom row) exhibits wide peaks, possibly encompassing multiple peaks. This inconsistency of ad hoc visual identification of peaks makes it subjective. Similarly, the topographic plots in *Appendix 4—figure 1A* demonstrates similar inconsistency: as estimation bandwidth increases (i.e., spectral resolution decreases), the topographic maps becomes less susceptible to spurious power leakage until the estimation bandwidth matches the bandwidth of underlying processes (compare middle row to top row). As the estimation bandwidth further increases, a severe spectral leakage renders the topographic maps largely uninformative of underlying separable oscillatory sources (bottom row).

ICA decomposition on this multichannel recordings identifies components that are neither slow oscillation nor alpha oscillation but combination of multiple oscillations as one would expect based on the observation that cumulative histograms of EEG recordings is approximately Gaussian (*Brookes et al., 2011*). We use 'extended infomax' (*Lee et al., 1999*) implementation provided by MNE-python 1.2 (*Gramfort et al., 2014*), the leading four ICA components are shown in *Appendix 4—figure 1B*.

Lastly, OCA is able to identify several oscillatory components (in both slow and alpha band) from this multichannel data, and extract their individual time series and topographic distribution. OCA is able to do so due to its explicit modeling of temporal dynamics in the form of the structured state-space representation. *Appendix 4—figure 1B* shows leading four components whose estimated center frequencies are within alpha band (9–13 Hz). This also demonstrates that OCA is bandwidth-free, that is, it adjusts estimation bandwidth to match the bandwidths of the underlying oscillations in a data-driven way as evident from the spectrum plots of the estimated oscillation time- courses.

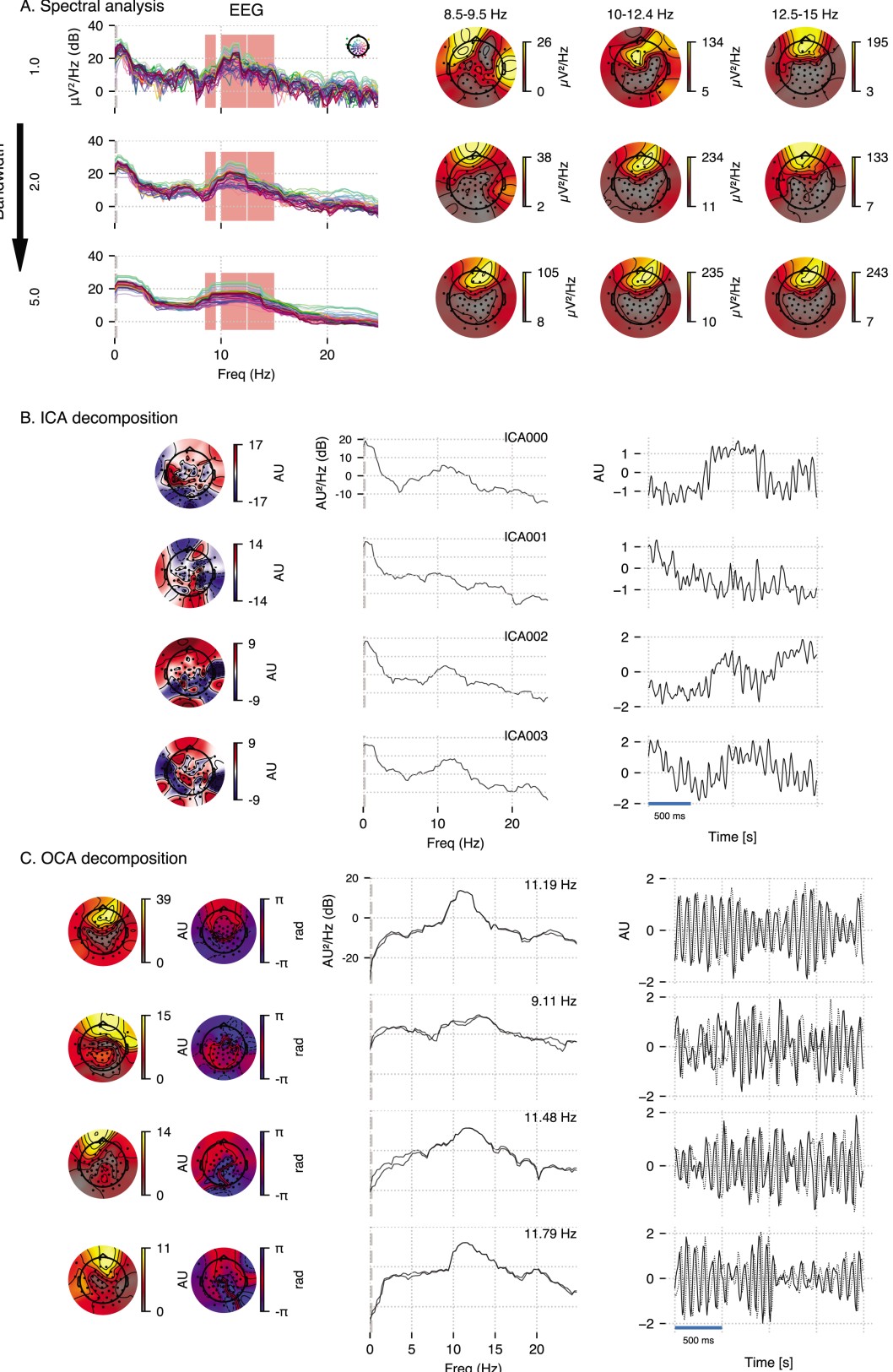

**Appendix 4—figure 1.** Empirical justification for oscillation component analysis (OCA) in analyzing real data. A 3.5-s-long electroencephalogram (EEG) recording during propofol-induced anesthesia (effect-site concentration of 2 μg) is considered for this demonstration. (**A**) Power spectral density using multitaper method (for varying time-
*Appendix 4—figure 1 continued on next page*

*Appendix 4—figure 1 continued*
bandwidth product) of the EEG recording and power distribution over the EEG sensors in the marked frequency bands (red overlay) around visually identifiable peaks. (**B**) Four leading independent component analysis (ICA) components (left, middle, and right columns show topographic maps, power spectrum density, and time courses, respectively). (**C**) Four leading OCA components within alpha band (the topographic maps show the magnitude [left] and phase [right], while line plots show power spectrum density [left] and time-courses [right], respectively).

