## [Editor Report · eLife assessment]

This method article proposes a **valuable** oscillation component analysis (OCA) approach, in analogy to independent component analysis (ICA), in which source separation is achieved through biophysically inspired generative modeling of neural oscillations. The empirical evidence justifying the approach's advantage is **solid**. This work will be of interest to researchers in the fields of cognitive neuroscience, neural oscillation, and MEG/EEG.

---

## [Referee Report · Reviewer #1 (Public Review)]

Summary:

The present paper introduces Oscillation Component Analysis (OCA), in analogy to ICA, where source separation is underpinned by a biophysically inspired generative model. It puts the emphasis on oscillations, which is a prominent characteristic of neurophysiological data.

Strengths:

Overall, I find the idea of disambiguating data-driven decompositions by adding biophysical constrains useful, interesting and worth pursuing. The model incorporates both a component modelling of oscillatory responses that is agnostic about the frequency content (e.g. doesn't need bandpass filtering or predefinition of bands) and a component to map between sensor and latent-space. I feel these elements can be useful in practice.

Weaknesses:

Lack of empirical support: I am missing empirical justification of the advantages that are theoretically claimed in the paper. I feel the method needs to be compared to existing alternatives.

Comments on the revised version: This concern has been addressed in the revised version.

---

## [Author Response]

The following is the authors’ response to the original reviews.

**Reviewer #1 (Public Review):**
Summary:The present paper introduces Oscillation Component Analysis (OCA), in analogy to ICA, where source separation is underpinned by a biophysically inspired generative model. It puts the emphasis on oscillations, which is a prominent characteristic of neurophysiological data.Strengths:Overall, I find the idea of disambiguating data-driven decompositions by adding biophysical constrains useful, interesting and worth-pursuing. The model incorporates both a component modelling of oscillatory responses that is agnostic about the frequency content (e.g., doesn’t need bandpass filtering or predefinition of bands) and a component to map between sensor and latent space. I feel these elements can be useful in practice.

Thank you for the positive evaluation!

Weaknesses:Lack of empirical support: I am missing empirical justification of the advantages that are theoretically claimed in the paper. I feel the method needs to be compared to existing alternatives.

Thank you for bringing up this important issue. We agree that a direct performance comparison would be important to demonstrate. We performed additional analyses to compare OCA with ICA and one easy frequency domain exploratory technique in both simulated and real human data (see Section How does OCA compare to conventional approaches? and Appendix 4: Comparison of OCA to traditional approaches in experimental EEG data). The results of the simulated data are shown in the revised Figure 3. Although the slow and alpha oscillations in this simulation are statistically independent under the generative model, ICA identifies components that mix these independent signals, as one would expect based on the above discussion (i.e., all components are Gaussian). Meanwhile, OCA is able to recover distinct slow and alpha components. We repeated this analysis in real human EEG during propofol-induced unconsciousness and found a similar result where ICA produced components that mixed slow and alpha band signals whereas OCA identified distinct oscillatory components (see Appendix 4—figure 1).

**Reviewer #1 (Recommendations For The Authors):**
MajorTheoretical justification. About the limitation of ICA In M/EEG, lines 24-28 seem to suggest that, almost by necessity (if Gaussianity approximately holds as argued), ICA doesn’t work on these modalities. But a body of work indicates that it does work to a reasonable extent, and that it is useful in practice; see https://www.pnas.org/doi/pdf/10.1073/pnas.1112685108?download=true. How then this theoretical claim be reconciled with the empirical evidence suggesting otherwise? I am putting this as a major comment because the limitations of ICA are one of the main motivations for this work, so it needs to be well-justified.

Thanks for bringing this forward this important point and for suggesting the reference Brookes, et al. Their work actually supports our claim. In the fifth paragraph of the discussion section, Brookes, et al. states “ICA has been used previously and extensively for artifact rejection in MEG; however, its use in identification of oscillatory signals has remained limited. This limitation is likely due to its susceptibility to interference and the fact that amplitude-modulated oscillatory signals exhibit a largely Gaussian statistical distribution (and ICA relies on non-Gaussianity in recovered sources).” For this reason, they use the Hilbert envelope as the input to the ICA procedure rather than the original time-series. These Hilbert envelopes represent the instantaneous amplitude of neural oscillatory activity, i.e., they follow the amplitude modulation of the oscillatory activity. The method does not extract any oscillatory activity or disambiguate different oscillatory sources, but only assess the connectivity pattern within pre-defined bands, i.e., how different areas of the brain are harmonized through modulation of the oscillations or vice-versa inside those pre-defined bands. The paper did not show extracted independent time signals (tICs), focusing instead on the spatial pattern that these tICs activated. In that way, their use of ICA was totally justified. Overall, our assessment of the limitations of ICA are very well aligned with Brookes, et al. We have added the against our claim in the introduction (see page 2, second paragraph) and revised the discussion section to refer to this paper (see page 15, second paragraph).

Empirical justification. The synthetic example is good, but I’m not quite sure what to make out of the real data examples. One can see reasonable spectra in the different bands and not-soeasy to interpret spatial topologies. But the main question is how OCA compares to more standard, easier approaches. Could the authors show explicitly how the benefits that were spelled out in the introduction/discussion manifest in practice, when compared to other methods?

Thank you for bringing up this important issue. We agree that a direct performance comparison would be important to demonstrate. We performed additional analyses to compare OCA with ICA and one easy frequency domain exploratory technique in both simulated and real human data (see Section How does OCA compare to conventional approaches? and Supporting Text: Comparison of OCA to traditional approaches in experimental EEG data). The results of the simulated data are shown in the revised Figure 3 in page 9. Although the slow and alpha oscillations in this simulation are statistically independent under the generative model, ICA identifies components that mix these independent signals, as one would expect based on the above discussion (i.e., all components are Gaussian). Meanwhile, OCA is able to recover distinct slow and alpha components. We repeated this analysis in real human EEG during propofol-induced unconsciousness and found a similar result where ICA produced components that mixed slow and alpha band signals whereas OCA identified distinct oscillatory components (see Figure Appendix 4—figure 1 in Appendix 4: Comparison of OCA to traditional approaches in experimental EEG data).

Minor"a recently-described class of state-space models" -> of the three references, one is from the sixties, another from the eighties, and the last one is 21 years old. Is this really a recent idea?Maybe rephrase "recently-described", or else think of more recent references that bring something new?

We have amended the wording as suggested. (See page 2, last paragraph)

Lines 72-74. It might be useful to unwrap in *intuitive* terms why the elements of this vector are closely related to the real and imaginary parts of the analytic signal.

Thanks for the helpful comment. The sentence now reads:

“These elements of this state vector traces out two time-series that maintains an approximate π/ 2 radian phase difference and therefore are closely related to the real and imaginary parts of an analytic signal…”. (See page 3)

Also, relatedly, I don’t seem to have access to the SI which is supposed to explain this. It doesn’t show up in the BiorXiv preprint either.

We are sorry to hear that. BiorXiv merges all the supporting information and posts them under the Supplementary Material.

In Eq(1) should it be R(f) instead of R(2 \pi f / f_s) ?

Thank you for catching this typo.

As I understand from lines 182-195, the input for the method is not channels but PCA components. Since R is learned, presumably the variance of the lower-order PCs (i.e. the latest elements of the diagonal of R) will estimated to be small. This, in turn, would make the likelihood to be heavily weighed on these components (because one basically divides their contribution by their variance). Would this potentially bias the estimation towards these lower-order PCs, at the expense of higher-order PCs. In a different context, this is shown here: https://journals.plos.org/ploscompbiol/article?id=10.1371/journal.pcbi.1008580 Maybe it would be worth commenting on this?

We agree with reviewer’s initial observations but disagree with the assessment. Our loglikelihood calculation reweights the components appropriately to counter the weighting coming due to spatial whitening, thus negating the above-mentioned bias. The main contribution of the spatial whitening and PCA are to make the learning numerically stable, i.e., it does not encounter underflow or overflow in the iterative steps. We also note that this spatial whitening, and the PCA are also reverted at the end to obtain the spatial components and estimated noise covariance. So, as long as we use all the components with strictly positive variances, we will not bias the log-likelihood one way or other.